# Reliability of vegetation resilience estimates depends on biomass density

Taylor Smith [1] ✉ & Niklas Boers[2,3,4]

Concerns have been raised that the resilience of vegetated ecosystems may be negatively impacted by ongoing anthropogenic climate and land-use change at the global scale. Several recent studies present global vegetation resilience trends based on satellite data using diverse methodological set-ups. Here, upon a systematic comparison of data sets, spatial and temporal pre-processing, and resilience estimation methods, we propose a methodology that avoids different biases present in previous results. Nevertheless, we find that resilience estimation using optical satellite vegetation data is broadly problematic in dense tropical and high-latitude boreal forests, regardless of the vegetation index chosen. However, for wide parts of the mid-latitudes—especially with low biomass density—resilience can be reliably estimated using several optical vegetation indices. We infer a spatially consistent global pattern of resilience gain and loss across vegetation indices, with more regions facing declining resilience, especially in Africa, Australia and central Asia.

Vegetated ecosystems worldwide are threatened by both intensifying land use and the growing impacts of anthropogenic climate change such as the increasing frequency and severity of droughts and heat waves[1]. Thoroughly monitoring the resilience of natural vegetation to changing shocks and stressors is therefore of crucial importance for anticipating and mitigating the impacts of ecosystem shifts, most importantly in terms of ecosystem services, food security and biodiversity loss[2].

A common approach to quantify the resilience of a given system is based on estimating its recovery rate from large perturbations[3]; the less resilient a system, the longer it will take to recover. A growing number of studies rely on satellite data to understand the climatic drivers of spatial variations in ecosystem resilience[4,5] and to quantify resilience changes through time[3,6–8] based on the concept of critical slowing down (CSD). CSD refers to the fact that, as a given system loses resilience, the restoring rate to its equilibrium state weakens, which can be measured in terms of rising variance and lag-one autocorrelation (AC1)[9–12]; the latter two CSD indicators have also been termed 'early-warning signals'[11]. The core of the CSD framework is that indicators such as AC1 or variance are theoretically related to the recovery rate of the system from large perturbations and in that sense to its resilience. CSD hence provides a theoretical framework to estimate resilience and variations thereof from time series data, even in cases without a catastrophic shift[13]; more direct ways to detect CSD—focusing directly on the restoring rate of the linearized dynamics—have also been proposed[3,14–16].

CSD has been applied to quantify changes in resilience or system stability in a wide range of contexts, including paleoclimate[10,17], present-day ice sheet dynamics[18,19], ocean circulation systems[15,20] and both global[3,6,8] and regional[7] vegetation systems. It should be emphasized, however, that CSD indicators are not the same as resilience but rather serve as proxies for resilience changes. In particular, there may in principle be other reasons for increasing variance or AC1 than CSD associated with resilience loss. This is why it is important to at least investigate both the AC1 and variance together and test whether their behaviour is consistent[21]. Note that, in situations with very high amplitudes of driving noise, CSD-based signs of resilience changes, or precursor signals of impending critical transitions, can be complemented or even replaced by investigations of so-called flickering, which arises when—in response to the strong noise forcing—the system begins to temporarily jump forth and back between alternative stable states[22].

[1]Institute of Geosciences, Universität Potsdam, Potsdam, Germany. [2]Earth System Modelling, School of Engineering and Design, Technical University of Munich, Munich, Germany. [3]Potsdam Institute for Climate Impact Research, Potsdam, Germany. [4]Department of Mathematics and Global Systems Institute, University of Exeter, Exeter, UK. ✉e-mail: tasmith@uni-potsdam.de

The theoretical relationships between the CSD indicators variance and AC1 and empirical estimates of the recovery rate after large perturbations have recently been confirmed for vegetation systems using global vegetation optical depth data[3]. However, the reliability of resilience estimates inferred from satellite vegetation data remains difficult to quantify, especially given the wide range of data sets available to monitor ecosystem health. The role of data aggregation (for example, spatial or temporal resolution) in biasing resilience estimates is also not well constrained. To obtain continuous records for longer periods, data are often constructed by combining signals from different satellites and sensors that were active across different time spans[23–25]. As different sensors generally have varying signal-to-noise ratios, this leads to non-stationary higher-order statistical characteristics even if means and trends are adjusted; this can therefore easily lead to spurious changes in CSD indicators that might be erroneously attributed to resilience changes[26].

Even with single-sensor products—for example, the range of vegetation indices provided by the Moderate Resolution Imaging Spectroradiometer (MODIS) instrument—it is not a priori clear whether the data capture the underlying vegetation dynamics sufficiently to be suitable for measuring vegetation resilience. Moreover, the effect of the level of temporal and spatial aggregation on the reliability of inferred resilience changes remains an open question. It has recently been shown that resilience estimates cannot easily be compared across land-cover types due to different baseline values for CSD-based resilience indicators[5]; the suitability and reliability of resilience indicators may also vary across vegetation types. Finally, in addition to using different data sets and resilience indicators, most studies quantifying vegetation resilience at regional to global scales do not agree on the specific methods used to pre-process satellite vegetation data before the analyses (for example, how to handle data gaps), which may lead to additional biases in CSD-based resilience indicators.

In this work, we first compare different data pre-processing methods used in recent studies quantifying vegetation resilience from satellite data by using synthetic data with known properties. This allows us to identify optimal choices for removing long-term nonlinear trends and seasonality, both of which can lead to biases in CSD-based resilience indicators. We then use Google Earth Engine[27] to process five MODIS vegetation indices—normalized difference vegetation index (NDVI), enhanced vegetation index (EVI), kernel NDVI (kNDVI)[28], gross primary productivity (GPP) and leaf area index (LAI)—at a range of spatial resolutions to examine the impact of spatial aggregation. To do so, we use the optimal deseasoning and detrending scheme identified with synthetic data in the first step, followed by the calculation of the restoring rate $\lambda$ from both the AC1 and variance. Theoretically, both AC1 and variance should lead to the same estimate of the recovery rate $\lambda$; deviations between the two estimates can thus be used to quantify the reliability of the corresponding resilience estimate. The translation of these methods to the Google Earth Engine[27] environment allows us to explore the suitability of different vegetation indices for measuring the resilience of vegetation systems globally at MODIS-native sensor resolution. We then quantify the impacts of the specific vegetation index, spatial aggregation and land-cover type on the reliability of resilience estimates at the global scale. Finally, we use only those locations where we find robust resilience estimates to explore recent trends in vegetation resilience.

## Comparing data processing schemes

To estimate the resilience of a given ecosystem via CSD, the time series encoding its dynamics must be approximately stationary—that is, long-term nonlinear trends and seasonal signals need to be carefully removed. There exist several methods to decompose a time series into its long-term nonlinear trend, seasonal and residual components[3,4,6,29], with commensurate strengths and weaknesses (Fig. 1).

While several recently used approaches remove the majority of the seasonal and long-term nonlinear trend, there remain key differences. First, removing a simple linear trend from data with a nonlinear trend does not yield a stationary time series; there remains a warping in the residual (Fig. 1e,f, black line). This is carried forward into time-explicit estimates of AC1; not only is the AC1 in this case higher than the 'perfect' baseline overall (Fig. 1g,h, blue line), but it also does not increase monotonically and shows considerable spurious variations towards the bifurcation-induced transition. Second, seasonal trend decomposition via Loess (STL) (Fig. 1g,h, purple line) has a similar AC1 slope as the perfect baseline when considering a simple seasonality model (STL slope = 0.9, perfect = 1.15) but overall lower AC1; we interpret this as STL overfitting the time series and thus removing too much of the actual signal. Finally, the rolling mean and harmonic fit approach very closely follows the perfect baseline in the simple example (harmonic slope = 1.08, perfect = 1.15) and is still closer to the perfect baseline than STL for the more complex example with variable seasonal timing and amplitude (Fig. 1b,d,f,h). Both the STL and the rolling mean detrender are unstable at the beginning and the end due to the well-known edge effects arising from rolling windows (that is, an incomplete number of data points are used to infer the slow nonlinear trend). We thus infer that, given a long data record (that is, where 2.5 year window can be discarded at the beginning and end for a 5 year rolling mean), the rolling mean followed by a harmonic fitting approach best reproduces the perfectly detrended and deseasoned data (Fig. 1g,h, red line).

It is important to note that optical remote sensing estimates of vegetation are highly dependent on surface cover. This means that data gaps are common, both over short (for example, clouds) and long (for example, winter snow) periods. Not all methods of removing trend and seasonality work equally well in the presence of gaps; for example, STL[29] was not originally designed to handle gaps. While there exist gap-aware implementations of the STL algorithm, it remains much more sensitive to gaps than a harmonic deseasoning approach. Regardless of the specific deseasoning method, it has been common practice to interpolate over or fill in missing data to create continuous time series. For example, ref. 6 used climatological means to gap-fill their vegetation data and ref. 3 used an upwards smoothing approach to interpolate over short gaps due to cloud cover. However, such gap filling will have knock-on effects on the stability of deseasoning and detrending methods and can easily induce biases in CSD-based resilience indicators[30]. In particular, gap filling based on climatology may lead to biases in variance and potentially AC1 if the distribution of gaps is not stationary. It is also not clear whether gap-filling is truly necessary—for a simple synthetic system (Fig. 1a) it can be shown that adding gaps of varying lengths does not bias the AC1 estimate systematically (Methods and Supplementary Figs. 1–3).

Adding longer and longer gaps (for example, up to 9 months of the year) increases the variability of the relationship between the AC1 of gappy and gap-free data but does not bias the relationship between the true AC1 and the AC1 inferred from the gappy data (Supplementary Fig. 2); this also holds true for the variance (Supplementary Fig. 3). Further, temporal resampling—as is often done to time-aggregate optical satellite data—decreases uncertainties in AC1 estimation and implies focusing on longer recovery timescales without biasing the AC1 estimates (Supplementary Fig. 4). We thus conclude that resilience can in fact be well constrained—given sufficiently long data records—without relying on any complex interpolation or gap-filling schemes that might subsequently bias resilience estimators.

From this we can draw two important conclusions: (1) our proposed deseasoning and detrending methodology is robust against data gaps, and (2) data gaps of varying sizes and frequencies found in the spatio-temporal field of real satellite vegetation data are unlikely to produce a systematic bias in CSD-based resilience indicators, at least over sufficiently long time windows. It is important to note that this is not necessarily true for all methods of deseasoning or detrending;

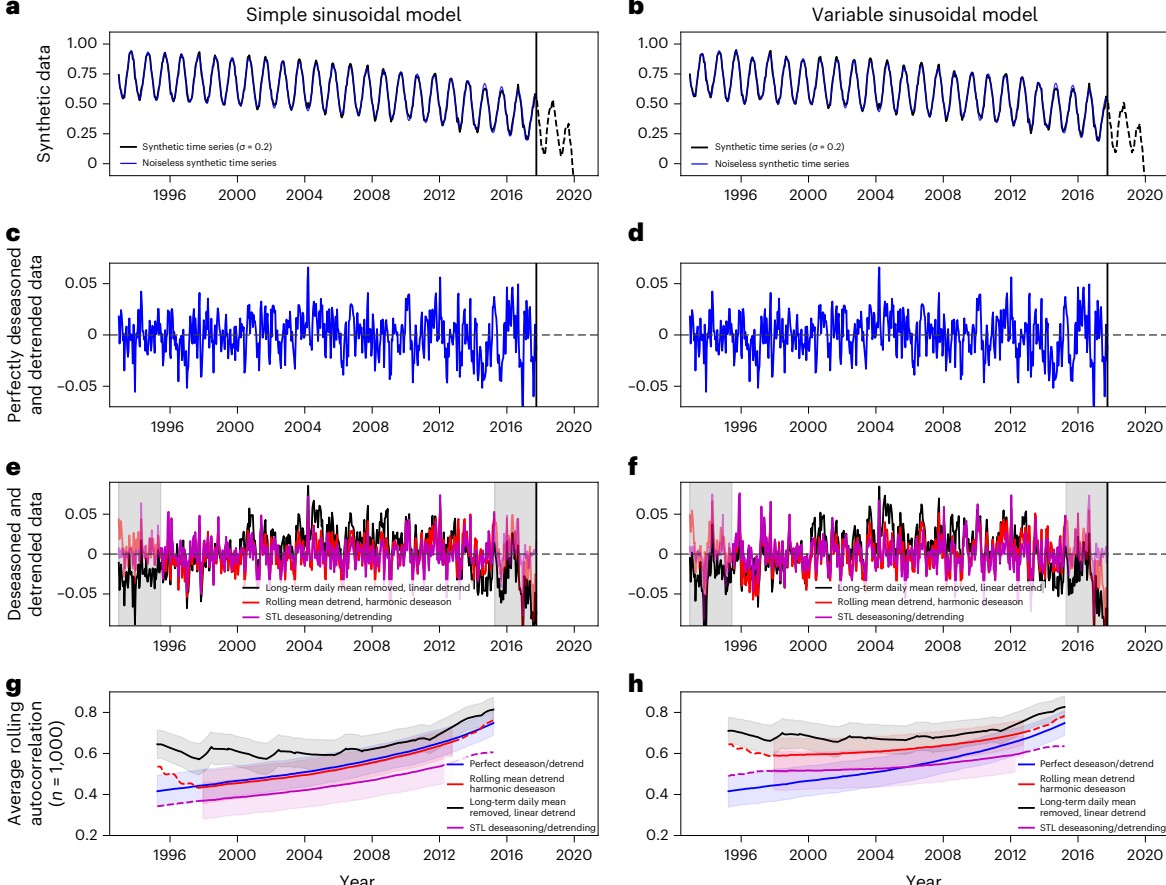

**Fig. 1 | Comparison of three standard deseasoning and detrending methods based on two paradigmatic time series examples showing resilience loss.** **a,c,e,g**, A simple sine curve is used for seasonality. **b,d,f,h**, A time- and amplitude-varying annual seasonality is used. **a,b**, Sample synthetic time series mimicking a seasonal vegetation curve (black line), moving towards a state transition (Methods). A version of the model without driving noise is plotted in blue. **c,d**, Stationary synthetic time series created by subtracting the noiseless model from the noisy model, that is, creating a 'perfectly' detrended and deseasoned residual. **e,f**, Resulting time series after applying standard statistical methods of deseasoning and detrending. Red, rolling mean then harmonic seasonality removal (Methods); black, remove long-term daily means, then fit a simple linear ramp detrender[6]; purple, remove trend and seasonality via STL[29]. Grey shading indicates the 2.5 year time spans at the beginning and end for which the rolling mean and STL residuals are unreliable because of edge effects inherent in rolling windows (that is, incomplete data windows in the beginning and end). **g,h**, Five-year rolling AC1 for each deseasoning/detrending method, showing that all methods correctly find overall increases, consistent with the model system approaching a transition, but with distinct differences in trend stability through time. Dashed lines indicate AC1 values based on residuals with less reliable detrending, that is, the first and last 2.5 years; shaded areas cover one standard deviation from the mean.

climatological means are sensitive to variable missing data (for example, if only one February during the full time series has data, all other February data will conform to that mean), and STL was not designed to concisely handle data gaps due to its nested local fitting approach.

## The reliability of resilience estimates

A system moving towards a bifurcation-induced transition will slow down critically; that is, the restoring forces that bring the system back to its equilibrium from continuous, small-scale and random disturbances become weaker and vanish at the critical transition point (Methods). This should be reflected by increases in both the AC1 and the variance; a trend in only one of these two parameters is not enough to confidently identify a change in resilience[3,21,26].

Based on the theory of CSD and Ornstein–Uhlenbeck processes, both AC1 and variance can be used to infer estimates of restoring rate $\lambda$; we will refer to them as $\lambda_{AC1}$ and $\lambda_{Var}$ in the following. Both estimates should be approximately equal if CSD is applicable and the restoring rate is to be interpreted in terms of resilience[12,15,21]. At the global scale, this relationship broadly holds; however, different land-cover types show widely varying $\lambda_{Var}/\lambda_{AC1}$ relationships (Fig. 2),

indicating that CSD is only appropriate to quantify resilience for certain vegetation types.

The degree to which different land-cover types follow the expected one-to-one relationship between $\lambda_{AC1}$ and $\lambda_{Var}$ is closely related to biomass—high-biomass regions tend to have lower correlations between $\lambda$ estimates (Fig. 2e and Extended Data Fig. 1). This pattern holds true for the different MODIS vegetation indices (Extended Data Fig. 2) and across spatial resolutions (Extended Data Fig. 3). Higher spatial resolution data tend to have more gaps overall (Supplementary Figs. 5 and 6); averaging over gappy or noisy data can improve the correlation between $\lambda_{AC1}$ and $\lambda_{Var}$, at the cost of reduced spatial resolution and potential mixing of disparate vegetation types within a single spatially aggregated pixel. For the examination of global-scale patterns, we choose 5 km data, which have also been used in recent publications[4,6] (Fig. 3). Regional- or local-scale analyses may find sensor-native (250 m and 500 m) data more appropriate in some contexts; these data can be easily produced with our methodology[31].

Our results show that CSD-based resilience estimates from MODIS vegetation indices (EVI, NDVI, kNDVI, GPP and LAI) are unreliable in many land covers—and especially in dense vegetation (Fig. 2 and

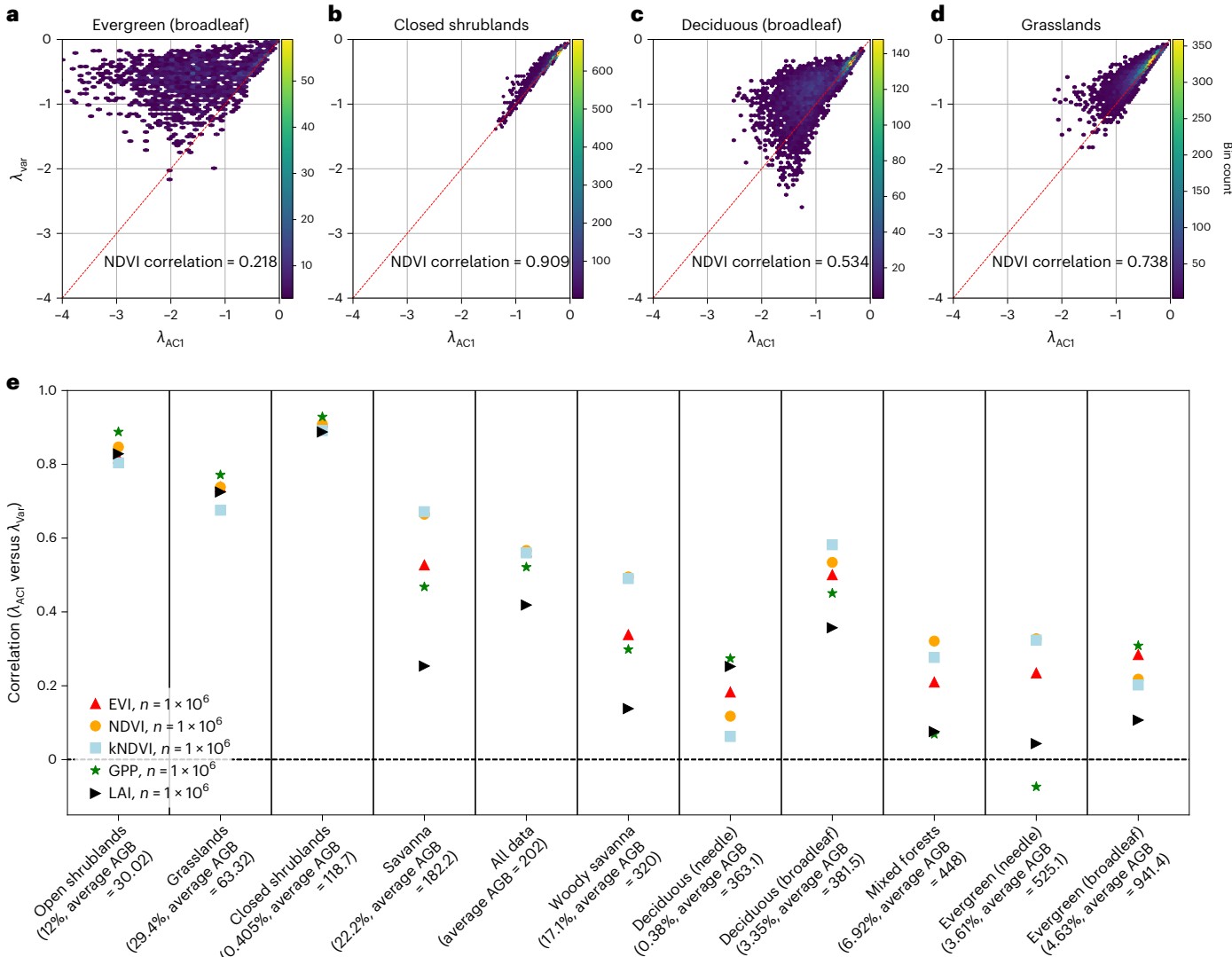

**Fig. 2 | Relationship between AC1- and variance-based estimates of the recovery rate λ using NDVI data at MODIS-native resolution (250 m).**
**a**–**d**, Evergreen broadleaf ($n = 10,000$) (**a**), closed shrublands ($n = 10,000$) (**b**), deciduous broadleaf ($n = 10,000$) (**c**) and grasslands ($n = 10,000$) (**d**) behave distinctly differently, with shrublands hewing closest to the expected one-to-one relationship between $\lambda_{AC1}$ and $\lambda_{Var}$ (red dashed line). See Extended Data Fig. 2 for other vegetation indices. **e**, Pearson's correlation for $n = 100,000$ points ($n = 10,000$ for each natural land-cover type), compared for all MODIS vegetation indices at native sensor resolution (EVI/NDVI/kNDVI, 250 m; GPP/LAI, 500 m) (see Extended Data Fig. 3 for a comparison of spatial resolutions). Land covers sorted by average above-ground biomass density (AGB)[47]. Global percentages (at 5 km resolution) of natural land-cover types on x-axis label. While most natural land-cover types have an overall positive correlation between $\lambda_{AC1}$ and $\lambda_{Var}$, some land-cover types follow the expected one-to-one relationship much more closely than others. CSD-based resilience estimation is problematic for land-cover types with lower correlation values.

Extended Data Figs. 1–3); hence, any inferences based on trends in AC1 or variance in those areas should be interpreted with caution (Fig. 3). Areas where $\lambda_{AC1}$ and $\lambda_{Var}$ do not agree broadly cluster in the tropics and the high northern latitudes (Fig. 3). Note the particularly low agreement between the two λ estimates for evergreen broadleaf forests (Figs. 2 and 3 and Extended Data Figs. 1–3); hence, especially for tropical rainforests, one should be careful when interpreting CSD indicators as reflecting resilience. For wide regions in the tropics, the theoretical formulae to infer λ from AC1 and variance yield undefined resilience estimates due to taking logarithms of negative values (Methods). It is well known that using NDVI over dense tropical forests is problematic due to NDVI saturation[32,33]; signal saturation damps variability and thus leads to biased estimates of vegetation dynamics. However, our results show that this problem for tropical forests is present not only for the NDVI but also for kNDVI, EVI and LAI, as well as GPP to a somewhat lesser degree (Extended Data Figs. 1–3

and Supplementary Fig. 7). While we use biomass here as an explanatory variable for poor resilience estimates, it is likely that other factors such as canopy closure could also explain this relationship; whenever the vegetation indices do a poor job of capturing ecosystem dynamics, resilience estimates will be less reliable.

To ensure that these inferences are robust across deseasoning approaches, we have repeated our analysis using STL (Supplementary Fig. 8). We find that the spatial patterns of λ estimates broadly agree, with the caveat that there are more undefined λ estimates when STL is used. This is due to the poor performance of STL when considering time series with a considerable number of gaps or too high noise levels; in many cases the STL fit results are too sparse to return a usable residual. Our detrending and deseasoning approach, based on rolling means and harmonic fitting, is somewhat more forgiving, but the overall spatial pattern of regions where signals are unreliable is very similar (compare Fig. 3 and Supplementary Fig. 8).

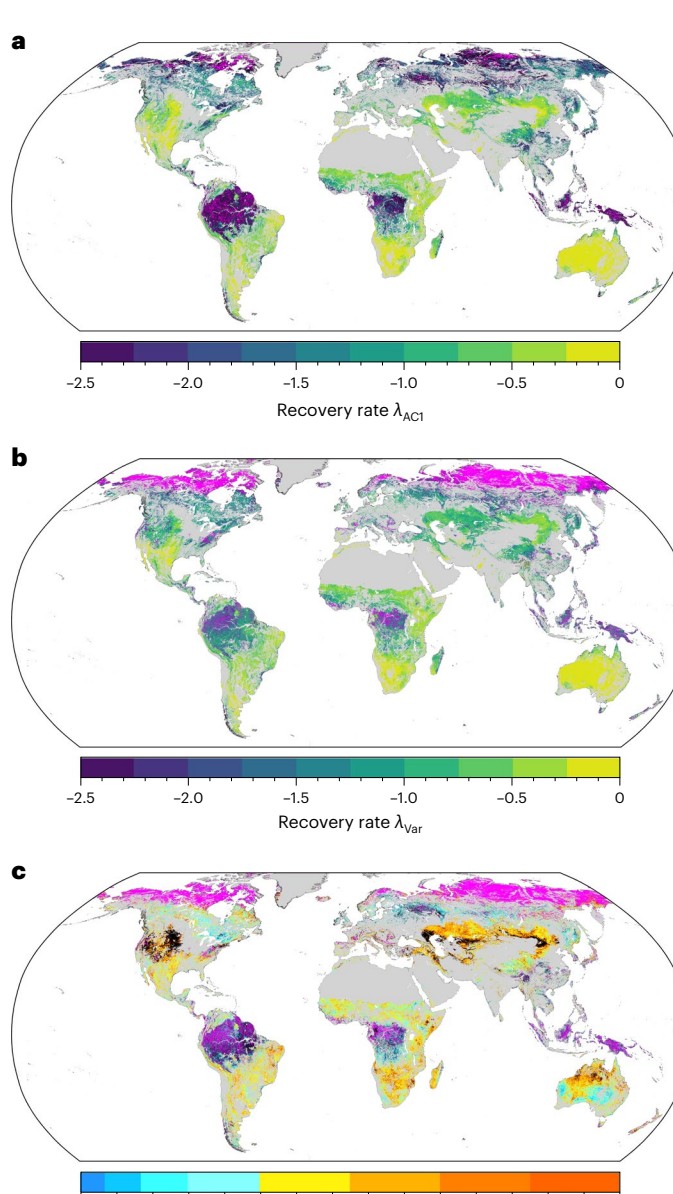

**Fig. 3 | Global patterns of the recovery rate $\lambda$. a,b**, AC1-based $\lambda$ (**a**) and variance-based $\lambda$ (**b**) for NDVI at 5 km resolution. Grey areas indicate land-cover types that are removed from our analysis due to human activity or absent vegetation (Methods), magenta areas indicate undefined estimates of the recovery rate, for example, due to negative AC1 values (Methods). **c**, Ratio of $\lambda_{Var}/\lambda_{AC1}$, again showing land cover mask (grey) and undefined $\lambda$ (magenta). Black areas indicate a large deviation (less than half or above a factor of 2) from the expected one-to-one relationship between $\lambda_{AC1}$ and $\lambda_{Var}$. Tropical rainforests and high-latitude boreal forest and tundra areas are unlikely to yield reliable estimates of changes in resilience based on MODIS vegetation indices. See Supplementary Fig. 7 for the same analysis using EVI, kNDVI, LAI and GPP, and Supplementary Fig. 8 for data processed with STL.

## Resilience trends

Despite the revealed limitations on resilience estimation at the global scale, there remains a large portion of the Earth's surface where estimates of resilience and its changes via CSD metrics—based on MODIS vegetation indices—are viable. Further, we have shown that our method of deseasoning and detrending allows us to robustly estimate resilience despite gaps in the time series (Supplementary Figs. 2–4). A changing

number or length of gaps would modify the variance and AC1 through time, potentially biasing any inferred changes in resilience[26]. However, we do not find a noticeable increase or decrease in the number of gaps over a 5 year window size; trends in the number of data points per window are limited to ~0.1 point per decade (Supplementary Fig. 9). Hence, a changing number of data points should not influence global-scale resilience change estimates.

We infer resilience changes—via both the AC1- and variance-based estimates of the recovery rate $\lambda$—using time windows of 5 years that are shifted by 1 year at a time (Methods). We limit this analysis to those areas where $\lambda_{Var}$ and $\lambda_{AC1}$ do not differ by more than a factor of 2; only where both $\lambda_{Var}$ and $\lambda_{AC1}$ have the same trend sign do we consider changes in resilience reliable (Fig. 4).

While large parts of the globe fail to produce reliable resilience estimates using any tested vegetation index (NDVI, kNDVI, EVI, GPP, LAI), there remain substantial regions where trends can be considered reliable. Eastern and southern Africa have large areas of reduced resilience across all indices, potentially driven by drying trends and changing human land use[34]; a multi-index resilience loss is also found in Australia (Extended Data Figs. 4–6). Central Asia shows consistent resilience loss across indices over a large area spanning from the Caspian sea to Mongolia, with the exception of a contiguous region in Kazakhstan showing resilience gains. The Americas show a more complex spatial pattern of resilience gains and losses (Fig. 4).

To take into account the combined information from the different vegetation indices, we compute the number of indices that agree on trend direction (Extended Data Fig. 6). We further confirm these results using STL to deseason the data (Supplementary Fig. 10); the spatial pattern of resilience gains and losses is similar, albeit with fewer inconsistent trends and more undefined trends than when deseasoning using a harmonic fitting approach.

## Discussion

MODIS data are widely used in studies of vegetation resilience[4–7,26,35–38]. For studying temporal trends in resilience—for example, in the context of the impacts of anthropogenic climate and land use change—the MODIS vegetation indices (EVI, NDVI, kNDVI, GPP, LAI) have the substantial advantage that they are single-sensor products; possible biases in resilience trends caused by the merging of signals from different sensors as, for example, for other NDVI[25], vegetation optical depth[23] or radar-based[24] data sets can hence be ruled out a priori[26]. However, systematic tests of the suitability of MODIS vegetation indices for resilience estimation are crucial, in particular regarding (1) specific detrending and deseasoning methods, (2) the role of spatial aggregation, (3) differences in baseline values of CSD-based resilience estimates (for example, due to plant physiology—grasses grow faster than trees), as well as in their reliability for specific land-cover types[5], and (4) differences in reliability of resilience estimates for different vegetation indices.

Our synthetic experiment (Fig. 1 and Supplementary Figs. 1–4) indicates that the specific means of detrending and deseasoning vegetation indices into stationary time series can exert strong effects on inferred resilience changes. Our proposed methodology is robust to multiple different seasonality models (Methods) and has the added benefit of working over data with gaps without the need for infilling those gaps which can, in turn, bias CSD indicators. For optical vegetation indices—like those from MODIS—gaps are common; a pre-processing method that does not rely on interpolation or infilling of climatology to handle data gaps hence minimizes potential induced biases in the resulting stationary time series[30].

We find—at the global scale—very similar results regarding CSD-based resilience estimators and their trends when using STL instead of the harmonic deseasoning approach (Supplementary Figs. 8 and 10). However, STL is much more sensitive to data with frequent or large gaps or data with high noise levels. In this way it may

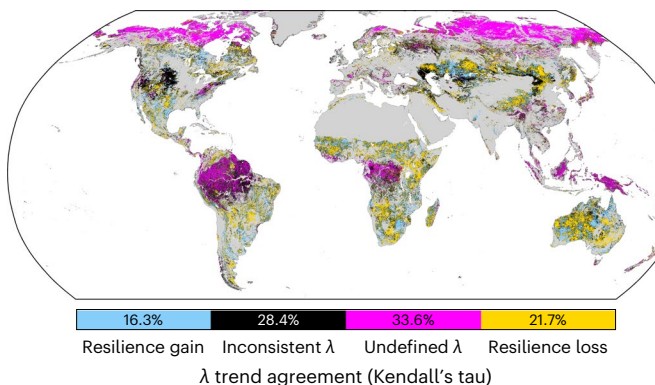

| 16.3% | 28.4% | 33.6% | 21.7% |
|---|---|---|---|
| Resilience gain | Inconsistent $\lambda$ | Undefined $\lambda$ | Resilience loss |

$\lambda$ trend agreement (Kendall's tau)

**Fig. 4 | Global patterns of resilience change in NDVI data, based on Kendall's tau trend agreement between $\lambda_{AC1}$ and $\lambda_{Var}$.** Grey areas indicate land-cover types that are removed from our analysis a priori (Methods), magenta areas indicate undefined $\lambda$ estimates (Methods), and black areas indicate a high $\lambda_{Var}/\lambda_{AC1}$ ratio (Fig. 3c) or disagreement in the trends of the two $\lambda$ estimates. Global patterns are broadly consistent across different vegetation indices, with a tendency towards loss of vegetation resilience globally (see Extended Data Fig. 4 for the same analysis using EVI, kNDVI, LAI and GPP, Extended Data Fig. 5 for the same analysis using linear trends and Supplementary Fig. 10 for data processed with STL).

be considered a more conservative approach; only well-behaved time series with minimal data gaps can be successfully processed with STL. Hence, the resilience trend maps produced using STL provide qualitatively similar outcomes but with fewer inconsistent trends and more undefined trends (Supplementary Fig. 10).

Satellite data are not always analysed at sensor-native resolution; in many cases some degree of spatial aggregation is desirable to reduce noise or ease processing constraints. We find that—in general—spatial aggregation does not strongly distort spatial patterns in the recovery rate (Extended Data Fig. 7). However, high spatial resolution data are more prone to data gaps (Supplementary Figs. 5 and 6) and have lower overall correlations between $\lambda_{AC1}$ and $\lambda_{Var}$ (Extended Data Fig. 3). This effect is particularly strong in dense vegetation (for example, rainforests) and relatively weaker in open vegetation landscapes (for example, grasslands, shrublands). We posit this is due to a mixing effect—aggregated time series will, all else being equal, be smoother due to the averaging out of independent noise and some data gaps and will hence have more stable residuals. There is thus a trade-off with spatial resolution—high spatial resolution has more uncertain time series but also less mixing of vegetation types. Globally, there is no one 'ideal' spatial resolution; the degree of aggregation required for $\lambda$ estimates to agree varies greatly.

We carefully excluded grid cells from our analysis where MODIS land-cover data indicate human activity, as well as any grid cells where the land-cover type has changed during the study period to ensure that our results, and especially the trends shown in Fig. 4, are not biased by anthropogenic land use change. Nevertheless, it should be noted that it is difficult to categorically exclude the possibility of anthropogenic influences, for example, under dense canopy cover where satellite-based estimates of human activity would be difficult.

Despite the robustness of our recovery rate estimates to both spatial resolution and data gaps, we find that MODIS vegetation indices are not appropriate for resilience estimation in all landscapes. In addition to the fact that vegetation types have different baseline values of CSD-based resilience estimates (AC1, variance or $\lambda$)[5], we have shown here that the reliability of CSD-based resilience estimates inferred from optical satellite vegetation indices varies strongly across land covers, with generally increasing difficulties for denser vegetation types (Fig. 2 and Extended Data Figs. 1–3). In particular, resilience estimates and changes thereof should be treated with caution in tropical rainforests,

where most of the AC1-based recovery rate estimates are undefined (Fig. 3 and Extended Data Fig. 7). These undefined estimates in the tropics are most common for NDVI, for which the susceptibility to saturation in high-biomass regions is well known[32,33]; surprisingly, however, recent improvements such as the EVI and kNDVI[28], and alternative metrics such as LAI, still show large and spatially coherent regions of undefined recovery rates in the tropics. Regions of undefined recovery rate in the densely vegetated tropical regions are smallest—but still considerable—for GPP (Supplementary Fig. 7). High-northern latitude boreal forests also produce undefined recovery rates, albeit primarily due to problems with $\lambda_{Var}$, rather than $\lambda_{AC1}$. This effect is likely due to short growing seasons followed by long periods of snow cover, which make $\lambda_{Var}$ estimates unstable (Methods). To confidently predict resilience and trends therein, both $\lambda_{AC1}$ and $\lambda_{Var}$ should be defined and furthermore should have similar values and trends (Methods).

We find that differences in the reliability of resilience estimates across land-cover types are broadly similar for each of the five MODIS vegetation indices considered here. As noted above, dense tropical forests and high-latitude regions including boreal forests present the greatest difficulties for resilience estimation. The general issues with NDVI saturation in tropical forests have been discussed widely[32,33]; any saturation will damp the dynamics measured by vegetation indices and hence lead to poor autocorrelation and variance estimates. We confirm that CSD-based resilience estimates perform poorly in high-biomass regions; our results imply that kNDVI and EVI are also impacted by saturation in high-biomass regions. GPP—while far from perfect—performs best in tropical forests and has the additional advantage of higher temporal resolution (8 days) compared to the EVI, NDVI and kNDVI indices (16 days). It should be emphasized that different indices perform better on different land covers; for example, EVI, NDVI and kNDVI outperform LAI and GPP in mixed and deciduous forests, as well as in woody and open savannas (Fig. 2). All five indices broadly perform better in low-biomass, 'open' landscapes such as shrub and grasslands than in high-biomass, 'closed' environments such as forests (Fig. 2). While further work is needed to fully constrain the reason for this difference, it is likely that optical vegetation indices do a better job of measuring ecosystem dynamics in low-biomass ecosystems.

The fact that CSD-based resilience estimation is particularly problematic in forests is unfortunate as both tropical and boreal forests have been suggested to be especially at risk of large-scale state transitions or even dieback in response to anthropogenic climate and land use change[7,37–40]. For these two crucial vegetation zones, other satellite vegetation data sets should generally be preferred. Given the issues in NDVI revealed here, we posit that Advanced Very High Resolution Radiometer NDVI data[25] would be particularly problematic for inferring resilience changes, as it suffers from both the problems related to NDVI itself and potential biases due to the merging of different sensors[26].

Nevertheless, we find that CSD-based resilience estimates are reliable in large parts of the world, especially in mid-latitude temperate and dryland environments with open vegetation cover (Fig. 3), and that there are many regions where trends agree across vegetation indices (Fig. 4 and Extended Data Figs. 4–6). Eastern and southern Africa, Australia and large parts of central Asia show a loss of resilience across all vegetation indices; however, parts of the southern United States and south-east Africa show a gain in resilience. Overall, losses in resilience outweigh gains; this finding is consistent across all MODIS vegetation indices. Based on our robustness checks, we conclude that resilience cannot be estimated reliably in some key regions, for example, in the higher northern latitudes or in the rainforests of the Amazon, Congo and Indonesia. In these regions, other vegetation metrics such as vegetation optical depth, which are less prone to saturation in dense vegetation[23], are likely to provide more reliable resilience estimates[3].

Our work shows that global-scale optical vegetation data can be used to measure vegetation resilience in land covers with more open vegetation cover—across spatial resolutions and despite data

gaps−given appropriate data preparation methods. We find that NDVI, EVI and kNDVI perform best for less densely vegetated landscapes such as grasslands. While many high-biomass, densely vegetated regions such as tropical rainforests or boreal forests do not provide robust estimates when using any optical vegetation indices, GPP slightly outperforms the other indices in these regions. In regions where we confirm the reliability of resilience estimates, we infer a tendency towards loss of resilience during the period 2000–2022; there exist spatially coherent regions of both increasing and decreasing vegetation resilience across all continents.

## Methods and data

### Satellite data

We use MODIS EVI and NDVI data (products MOD13Q1 (250 m)[41] and MOD13A2 (1 km)[42], 2000–2022, 16 day composites), as well as GPP (MOD17A2, 500 m, 2000–2022, 8 day composites[43]) and LAI (MCD15A3H, 500 m, 2002–2022, 4 day composites[44]) to examine global vegetation dynamics. All vegetation data sets are available via Google Earth Engine[27]; we further generate our own 1 km, 5 km, 10 km and 25 km resolution products via spatial averaging. We only use data points flagged as 'highest quality' in our analysis. Finally, we also generate the recently introduced kNDVI metric[28] for completeness. In contrast to previous publications[3,6], we do not gap-fill our vegetation data.

We use MODIS land-cover data (MCD12Q1, 500 m, 2001–2021[45]) to both mask out non-vegetated areas (for example, urban areas) and to subdivide our results by land-cover type. We further mask out any land covers that have changed (for example, forest to agriculture) during the period 2001–2021 to limit the influence of ecosystem transitions or anthropogenic influence on our results. Land cover data at 250 m resolution uses a nearest-neighbour resampling; 1 km, 5 km, 10 km and 25 km data use the mode of input land covers. To minimize the impact of anthropogenic and changed land cover, we further remove any pixels which have more than 10% of their area masked out (for example, a pixel that is 89% forest and 11% urban or cropland is removed). Python code to reproduce our land-cover masking, data pre-processing and data exports can be found on Zenodo[31].

To create Fig. 2, we use a stratified random sample of 100,000 locations covering the 10 relevant natural land-cover types with an equal number of samples (International Geosphere-Biosphere Programme type 1[45]). Sample locations and script used to generate the random samples and export the data can be found on Zenodo[31]. A secondary sampling scheme based on 100,000 random points distributed evenly between World Wildlife Fund Ecoregions[46] yielded similar results and is thus not shown here. Finally, we use a global above-ground biomass density estimate (2010 composite[47]) to assess how the reliability of the recovery rate λ depends on biomass, averaged by land-cover type (Fig. 2 and Extended Data Figs. 1 and 3).

### Synthetic time series

To estimate resilience using CSD, the time series in question must be approximately stationary, that is, without long-term (nonlinear) trends and seasonality. To provide an initial comparison of deseasoning and detrending methods, we create synthetic time series $X(t)$ for which the ground truth is known by numerically integrating a paradigmatic example of a 'double-well' dynamical system that shows bistability for a certain parameter range and bifurcation-induced transitions between the two alternative stable states as the control parameter is varied, namely,

$$dX_t = (-X_t^3 + X_t - p)dt + \sigma dW \tag{1}$$

where $X_t$ denotes the system state at time $t$, $p$ is the control parameter which is gradually varied from −1 to +1 to produce a bifurcation-induced transition, $W$ denotes a Wiener process that is used as the noise driving the system, and $\sigma$ is the amplitude of that noise. We simulate these time series over a period of 31 years, using a daily time interval.

We introduce seasonality in a variety of ways to explore its influence on the different deseasoning procedures. In the simplest case (Fig. 1, left column), we use a stable sine curve for seasonality; we also perform a simulation where the seasonal amplitude and timing is randomized year-on-year (Fig. 1a,c,e,g). For further robustness checks, we also include seasonal models with separate time- or amplitude-randomization, additional noise and multiple annual peaks (Supplementary Figs. 11 and 12). Python code to reproduce our synthetic time series can be found on Zenodo[31].

### Deseasoning and detrending

A wide body of literature discussing the best way to deseason and detrend time series exists; we do not focus here on an exhaustive inter-comparison of these methods. We tested three common deseasoning and detrending schemes on our synthetic data: (1) a rolling mean detrender followed by removing a third-order harmonic function fit to the data as a deseasoner, (2) removing long-term daily means, followed by a simple linear detrender[6,8] and (3) STL[3,29]. We compare the performance of these methods on synthetic data with a 'perfect' deseasoner, which is created by running our synthetic model without the additional white noise, that is, $\sigma = 0$ and no added sine curve. Hence, we can create a control data set by perfectly removing the long-term drift and seasonal components, leaving only the stationary residual of interest for the CSD analysis (Fig. 1c,d). We note that other schemes−for example, using a rolling linear fit instead of a rolling mean, or using higher- or lower-order harmonic fits to deseason−perform similarly; to simplify our discussion, we only highlight these three methods. We compare the performance of these schemes by computing the 5 year rolling AC1 over 1,000 simulations and taking the average AC1 at each time point (Fig. 1g,h).

We find that removing long-term daily means and a linear detrender performs poorly across all model simulations (Fig. 1 and Supplementary Figs. 11 and 12); the resultant AC1 time series has several spurious jumps even for simple constant sinusoidal seasonality. STL performs well for simple or randomly varying seasonality; it does, however, tend to produce lower AC1 estimates than expected. We posit this is due to overfitting, where some random noise is incorrectly placed in the seasonality rather than the residual component.

Overall, our results suggest that a rolling-mean detrender followed by a harmonic fit deseasoner performs best when compared to the true underlying detrended and deseasoned data. As we perform a single harmonic fit to each time series, the long-term average seasonality will be removed from the time series. If seasonality changes randomly through time, there will not be a bias in the AC1 calculated over the residual. If the seasonal amplitudes increase monotonically through time, a trace of this may be left in the residual that could affect the CSD-based resilience indicators. In such cases, one would need to decide whether the monotonic changes in seasonality are part of the climatic forcing or of the vegetation response. In the first case, a deseasoning method designed for variable seasonal amplitudes would likely be preferable. In the latter case, the changing response to a constant climate forcing could be indicative of resilience changes and should be kept in the residual; in such cases, deseasoning via harmonic fitting over the entire time span should be preferred. As interpolation or gap-filling can bias CSD-based resilience metrics[30], a key advantage of a harmonic deseasoner is that it is substantially less sensitive to data gaps than STL (Fig. 3 and Supplementary Fig. 8).

### Data gaps and spatial resolution

At very fine spatial scales (that is, 250 m), gaps in the different MODIS vegetation indices are very common; it is only when these data are aggregated to much larger spatial scales (we compared 1 km, 5 km, 10 km and 25 km for EVI, NDVI and kNDVI) that quasi-continuous time series are created (Supplementary Figs. 5 and 6). To test the impact of

variable data gaps on our long-term estimates of resilience, we return to our synthetic model (Fig. 1a). We first remove a random sample of data to simulate variable cloud-cover gaps and then remove increasingly long time periods to simulate summer or winter gaps, especially due to snow cover (see Supplementary Fig. 1 for an example time series). We also resample our data to a bi-weekly average; this is commonly done with satellite vegetation data to deal with cloud gaps or other missing data. We then calculate AC1 using a rolling 5 year window over the ~25 years of data.

Overall, the relationship between the AC1 of the gap-free and gappy time series is very close to being one-to-one; this holds true even for long gaps (Supplementary Figs. 1–3). For a higher number of gaps, variability around the one-to-one line increases, implying less certain AC1 estimates; however, the temporal resampling suppresses this variability and hence leads to more certain AC1 estimates (Supplementary Fig. 4).

For empirical vegetation data, we infer that within the reasonable bounds of 250 m and 25 km, our results remain qualitatively similar (Extended Data Fig. 3); however, in particular for the tropics, as expected there are fewer regions with undefined recovery rate estimates for coarser resolution due to higher amounts of spatial aggregation (Extended Data Fig. 7).

## Estimating vegetation resilience

Concerning resilience estimates, the fundamental quantity we are interested in is the recovery rate from perturbations. It can be shown that this is the same as the restoring rate $\lambda$ of a linearized version of the dynamics around a given equilibrium[3,12], which is technically given by an Ornstein–Uhlenbeck process[48]:

$$dX_t = \lambda X_t dt + \sigma dW \tag{2}$$

for deviations $X_t$ from the equilibrium, where $\lambda < 0$ for stable dynamics, and increasing (decreasing) recovery rate $\lambda$ indicates a loss (gain) of resilience; note that $\lambda$ approaches 0 from below as the bifurcation points of the double-well system above (Fig. 1) are approached—this is the key characteristic of CSD. In order for the theory to be applicable in practice, the above Ornstein–Uhlenbeck process has to be discretized into equal time steps of size $\Delta t$ (which we set to 1 for simplicity), which yields the characteristic order-one auto-regressive process[49]:

$$X_{n+1} = aX_n + \tilde{\sigma}\eta_n \tag{3}$$

where $X_n$ denotes the system state at discrete time step $n$ and $\eta_n$ is independent normally distributed white noise. Note that based on this equation, the autocorrelation $a$ and $\tilde{\sigma}^2$ can be inferred from empirical time series by regressing $X_{n+1}$ onto $X_n$.

It can be shown that the autocorrelation at lag $n$ is given by[3,49]

$$\alpha(n) = e^{\lambda n \Delta t} \tag{4}$$

where $e$ is the exponential function, and thus in particular $\alpha(1) = a = e^{\lambda \Delta t}$. The variance of the discrete driving noise $\tilde{\sigma}^2$ is given by

$$\tilde{\sigma}^2 = -\frac{\sigma^2}{2\lambda}\left(1 - e^{2\lambda \Delta t}\right). \tag{5}$$

The variance of the full discretized time series can then be shown to be[3,15,18]

$$\text{Var}[X] = \frac{\tilde{\sigma}^2}{1 - e^{2\lambda \Delta t}} = -\frac{\sigma^2}{2\lambda} \tag{6}$$

where we used the above identity for $\tilde{\sigma}^2$ in the second equality.

Based on the above, we directly find an estimate of the recovery rate based on the AC1 via

$$\lambda_{\text{AC1}} = \frac{1}{\Delta t}\log(a) \tag{7}$$

Similarly, we can infer a second estimate of the recovery rate from the variance, namely,

$$\lambda_{\text{Var}} = \frac{1}{2\Delta t}\log\left(1 - \frac{\tilde{\sigma}^2}{\text{Var}[X]}\right). \tag{8}$$

Note that, importantly, these two equations for $\lambda_{\text{AC1}}$ and $\lambda_{\text{Var}}$ only contain quantities that can be directly inferred from a linear regression of $X_{n+1}$ onto $X_n$ for empirical time series $X$. It is clear that, if the theoretical conditions underlying the CSD theory hold, the two estimates for the recovery rate should agree. Deviations from a one-to-one relationship can therefore be used as a metric for the suitability of a given time series to be used for CSD-based resilience estimation (Figs. 2 and 3).

The two recovery rate estimates $\lambda_{\text{AC1}}$ and $\lambda_{\text{Var}}$ are computed pixel-wise globally over the entire deseasoned/detrended time series for each of our EVI, NDVI, kNDVI, LAI and GPP data sets using the Google Earth Engine platform[27] (code repository on Zenodo[31]). As a logarithm is present in both equations for the recovery rate estimates, $\lambda$ cannot be inferred for regions where either $a < 0$ (for AC1) or $\frac{\tilde{\sigma}^2}{\text{Var}[X]} > 1$ (for variance). For the AC1, these regions are mostly found in the tropics (Fig. 3), where signal saturation and noise reduce the AC1 (for example, Supplementary Fig. 7). In large parts of the high northern latitudes, large values of $\tilde{\sigma}$ lead to a negative argument of the logarithm and thus undefined estimates of $\lambda_{\text{Var}}$. We posit that this is driven by short growing seasons, which increase $\tilde{\sigma}$ while reducing Var[$X$], leading to unconstrained $\lambda$ estimates.

## Defining trends in resilience

To calculate trends in resilience, we construct overlapping 5 year windows between 2002 and 2020 to ensure that all windows have roughly the same number of data points. We then count the number of data points in each window and estimate both $\lambda_{\text{AC1}}$ and $\lambda_{\text{Var}}$ for each pixel that is part of the analysis (excluding, for example, human-affected regions as described above) from the deseasoned and detrended data. We repeat this analysis for the different considered vegetation indices (EVI, NDVI, kNDVI, LAI, GPP) at 5 km spatial resolution. Trends in the resilience indicators are estimated via both Kendall's tau statistics (Extended Data Fig. 4) and the slope of a simple linear regression (Extended Data Fig. 5). We only compare trend direction, not magnitude, to give a general picture of where resilience change estimates can be considered reliable (Fig. 4). Due to the well-known edge effects of rolling averages (Fig. 1g,h), we also check our trends over only the middle period of our data (2004–2017). We find broadly similar spatial patterns globally, albeit with slight shifts regionally (Supplementary Fig. 13); we also find that the spatial pattern of trends is very similar when data are pre-processed using STL instead of harmonic deseasoning (Supplementary Fig. 10).

Moreover, we test whether the size of gaps in the data sets has changed through time and find that changes in the density of data gaps in each window are negligible. In general, the amount of data has increased by one measurement per decade (that is, 0.5 measurements per window) in most of the world (Supplementary Fig. 9). We thus conclude that changes in data gaps do not have an outsize influence on our estimated resilience trends.

## Reporting summary

Further information on research design is available in the Nature Portfolio Reporting Summary linked to this article.

## Data availability

The satellite data used in this study is publicly available[41–45] and can be accessed offline or via Google Earth Engine[27]. Synthetic data can be reproduced via codes available on Zenodo[31]. The 100,000 random sample locations used in Fig. 2 are also available via Zenodo[31].

## Code availability

Python scripts to deseason/detrend and export MODIS vegetation data, as well as code to reproduce the synthetic data used in this study, can be found on Zenodo[31].

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

## Acknowledgements

The State of Brandenburg (Germany) through the Ministry of Science and Education and the NEXUS project supported T.S. for part of this study. T.S also acknowledges support from the BMBF ORYCS project, the DFG STRIVE project (SM 710/2-1), and the Universität Potsdam Remote Sensing Computational Cluster. N.B. acknowledges funding from the Volkswagen Stiftung, the European Union's Horizon 2020 research and innovation programme under grant agreement number 820970 and under the Marie Sklodowska-Curie grant agreement number 956170, as well as from the Federal Ministry of Education and Research under grant number 01LS2001A. This is TiPES contribution #244.

## Author contributions

T.S. and N.B. conceived and designed the study and interpreted the results. T.S. processed the data and performed the numerical analysis. T.S. and N.B. wrote the paper.

## Funding

## Competing interests

The authors declare no competing interests.

## Additional information

**Extended data** is available for this paper at https://doi.org/10.1038/s41559-023-02194-7.

**Correspondence and requests for materials** should be addressed to Taylor Smith.

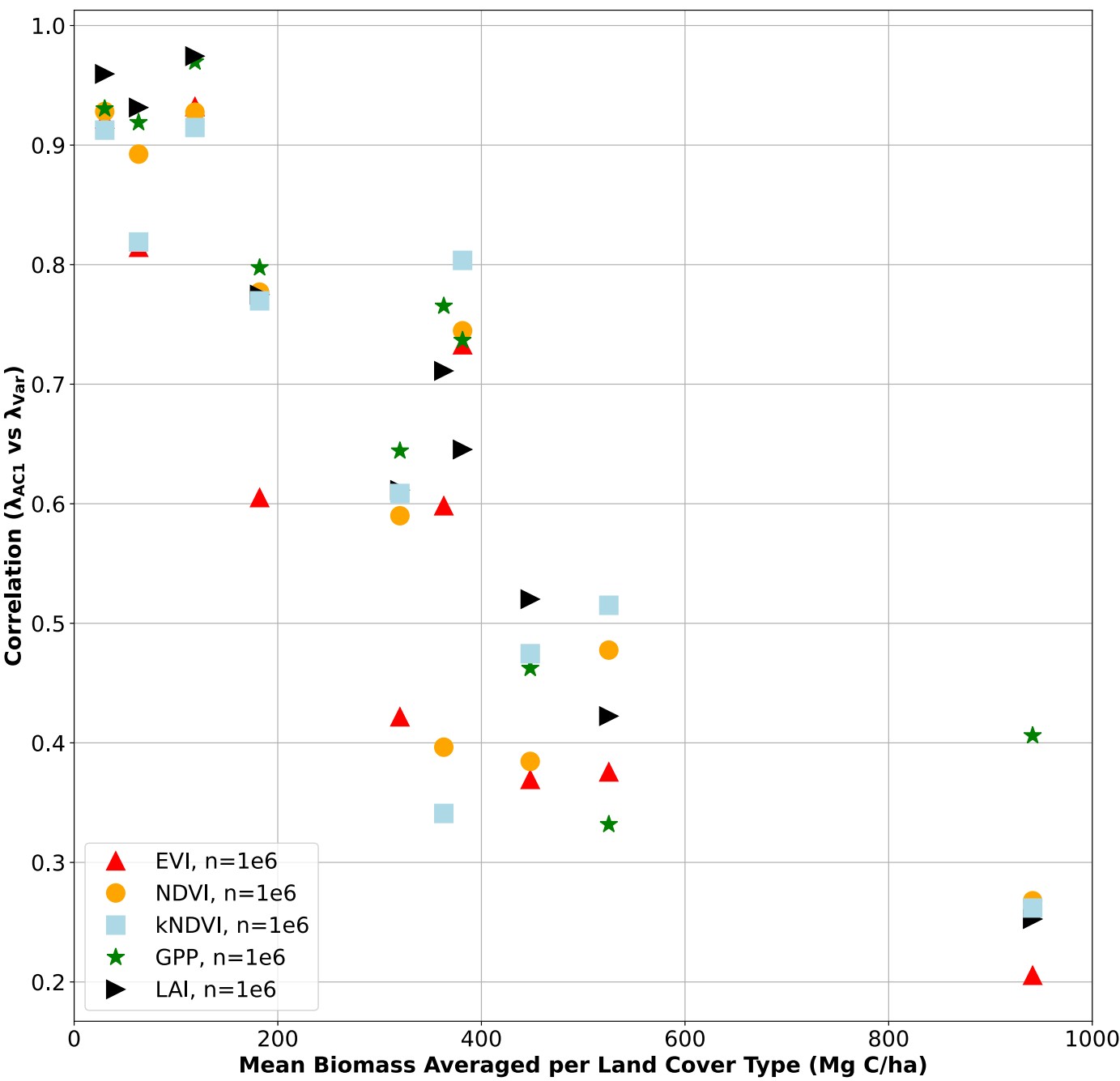

**Extended Data Fig. 1 | Comparison between biomass and $\lambda_{Var}/\lambda_{AC1}$ correlation coefficients.** The floating x-scale emphasizes the strongly linear relationship between correlation and biomass across all indices; n=100,000 for each index, n=10,000 per land cover type.

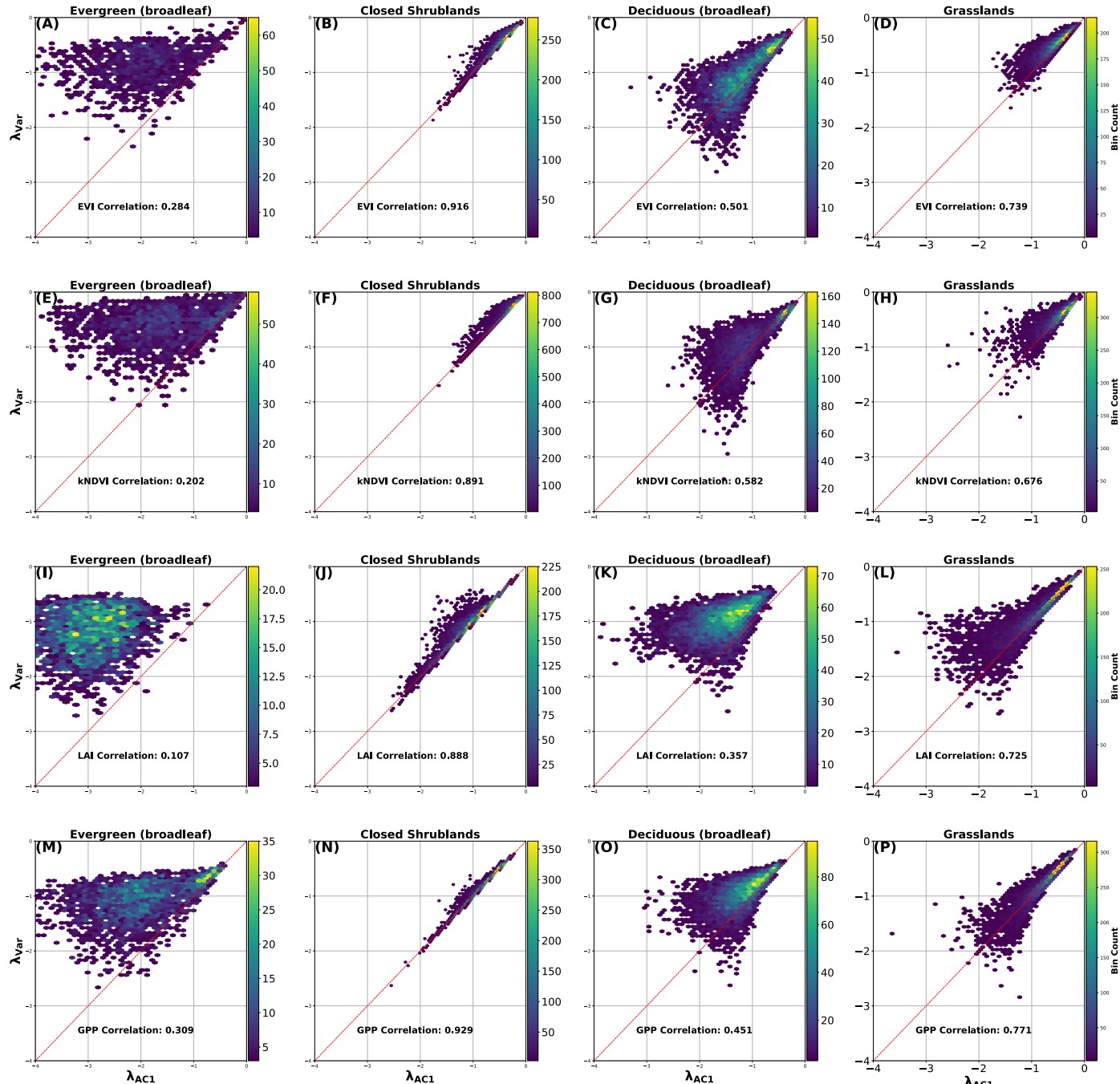

**Extended Data Fig. 2 | Comparison of $\lambda_{Var}/\lambda_{AC1}$ correlation by land cover type.** Each row covers one vegetation index (n=10,000 per land cover type), from top to bottom: EVI (A,B,C,D), kNDVI (E,F,G,H), LAI (I,J,K,L), GPP (M,N,O,P). Correlation coefficients listed on charts, with red 1:1 line shown for reference. Minimum 3 points per bin.

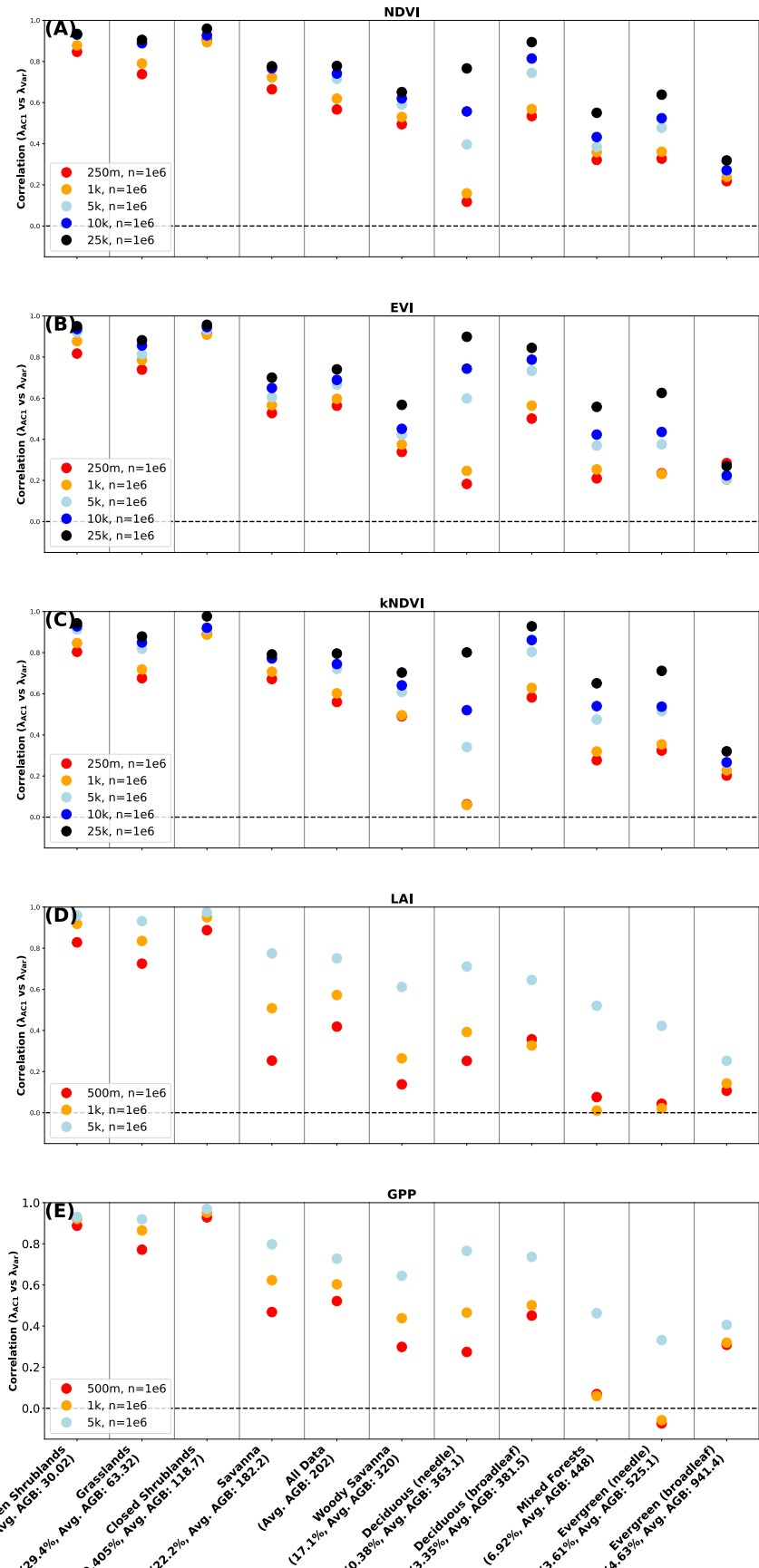

**Extended Data Fig. 3 | Comparison of $\lambda_{Var}/\lambda_{ACI}$ correlation by spatial aggregation.** Each row covers one vegetation index (n=100,000, n=10,000 per land cover type). (A) NDVI, (B) EVI, (C) kNDVI, (D) LAI, (E) GPP. Land cover types sorted by above-ground biomass (AGB). 10 km and 25 km data not included for LAI/GPP due to processing constraints.

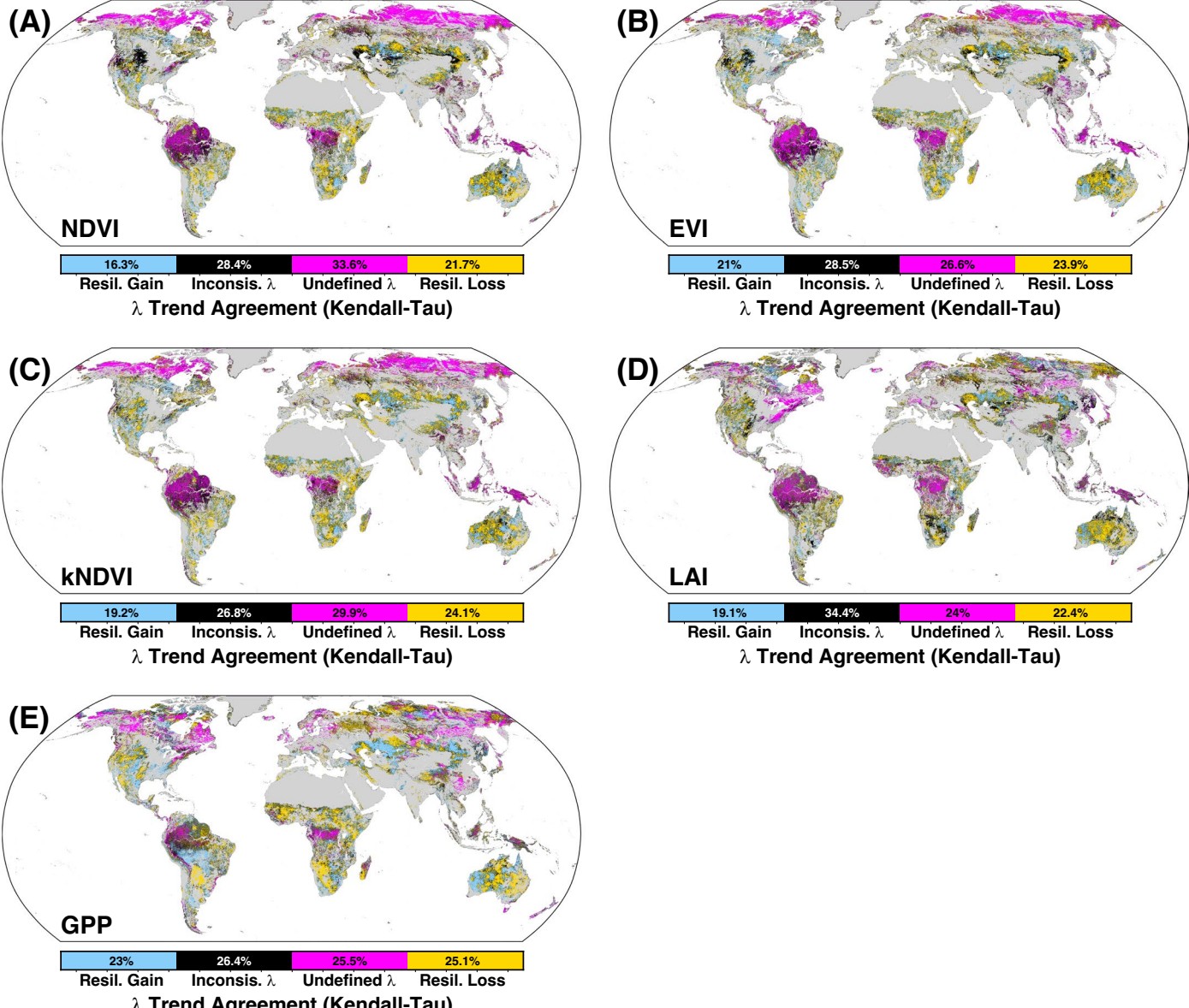

**Extended Data Fig. 4 | Global Kendall-Tau trends in resilience across all vegetation indices at 5 km resolution.** (A) NDVI, (B) EVI, (C) kNDVI, (D) LAI, (E), GPP. Grey areas masked for land cover (see Methods). Areas of agreement between variance- and AC1-based $\lambda$ marked as resilience gain or loss, others as inconsistent (high $\lambda_{var}/\lambda_{AC1}$ ratio or trend disagreement, black) or undefined $\lambda$ (magenta).

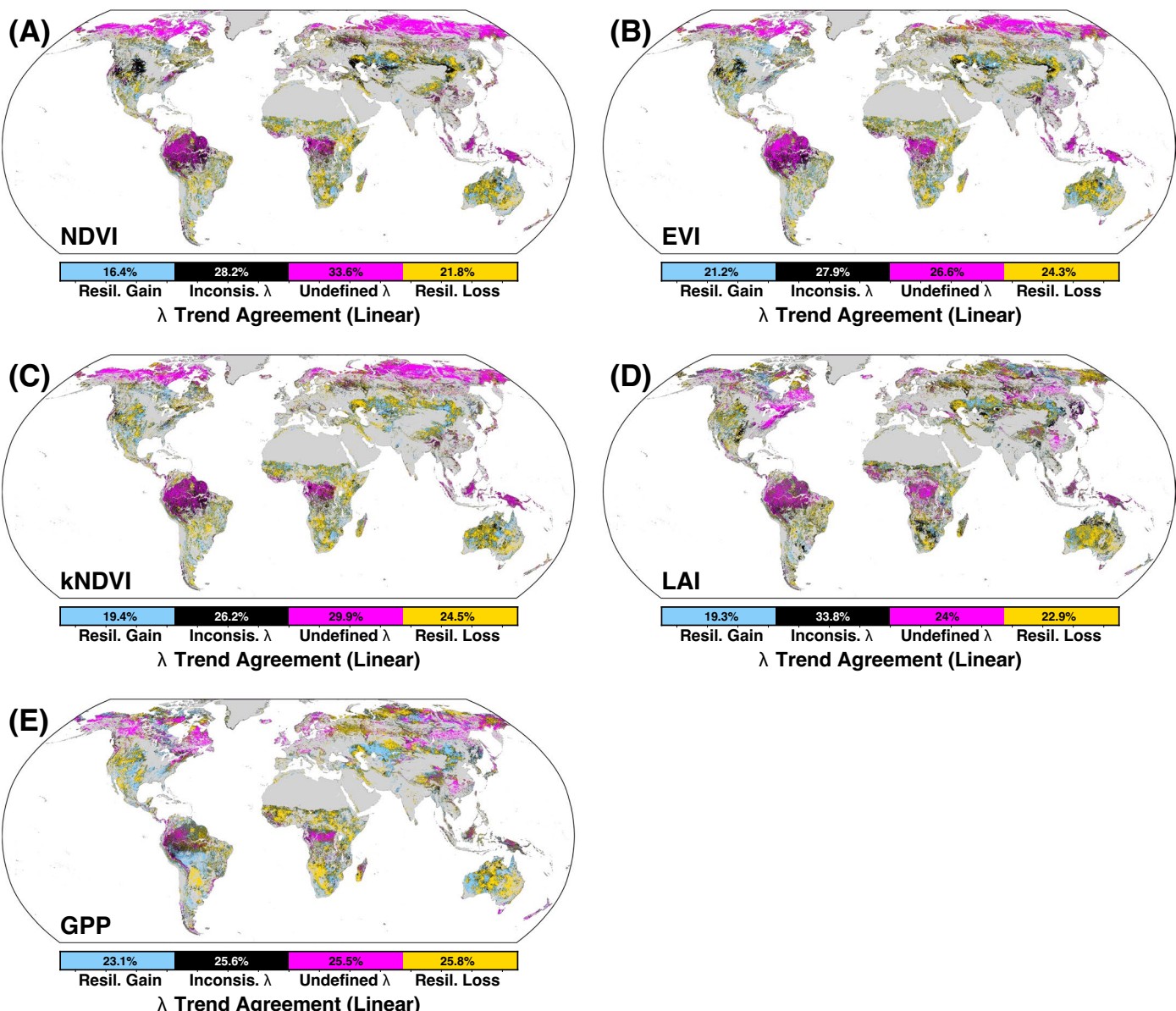

**Extended Data Fig. 5 | Global linear trends in resilience across all vegetation indices at 5 km resolution.** (A) NDVI, (B) EVI, (C) kNDVI, (D) LAI, (E), GPP. Grey areas masked for land cover (see Methods). Areas of agreement between variance- and AC1-based $\lambda$ marked as resilience gain or loss, others as inconsistent (high $\lambda_{Var}/\lambda_{AC1}$ ratio or trend disagreement, black) or undefined $\lambda$ (magenta).

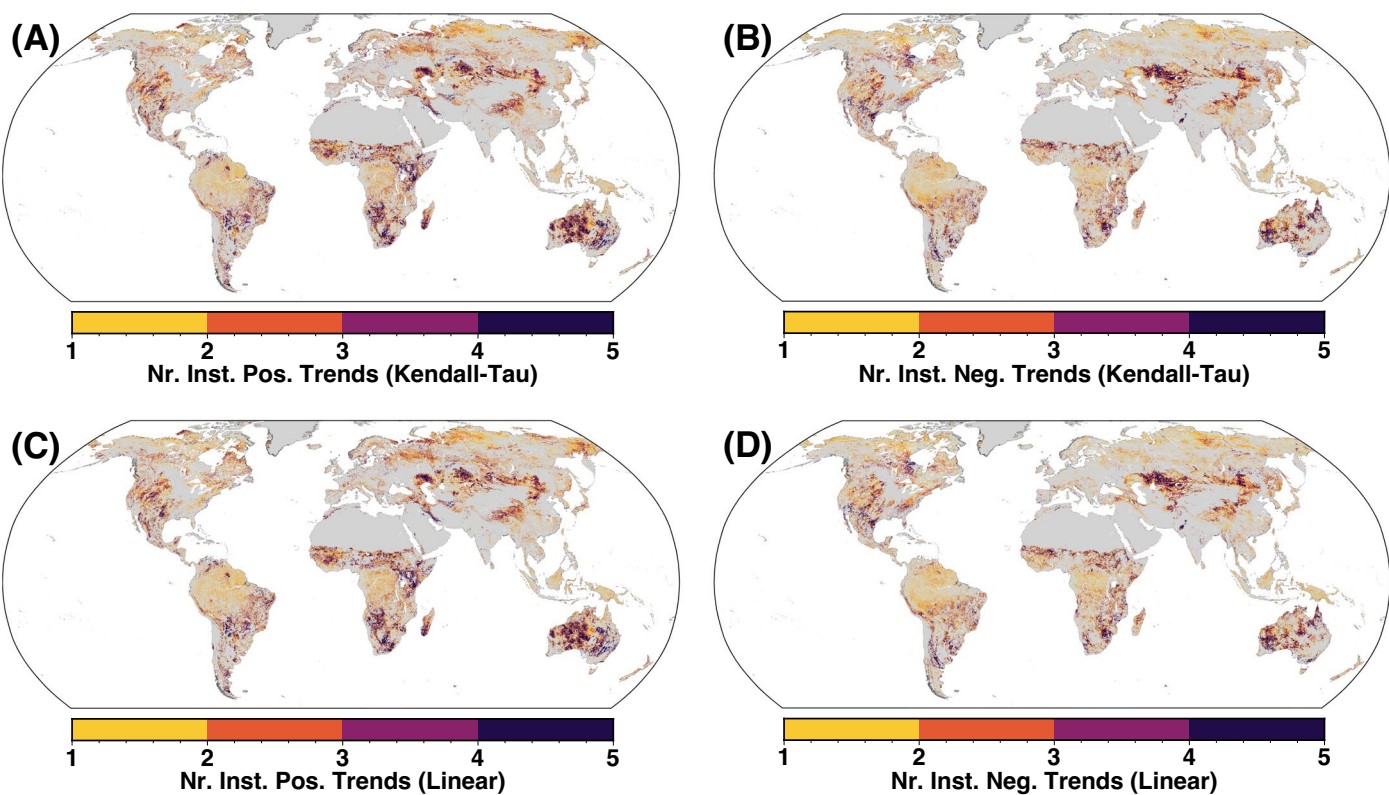

**Extended Data Fig. 6 | Number of instruments agreeing on the direction of resilience trend (positive/negative).** (A,B) Kendall's tau statistics and (C,D) linear trends. Grey areas masked out due to anthropogenic or vegetation-free land cover.

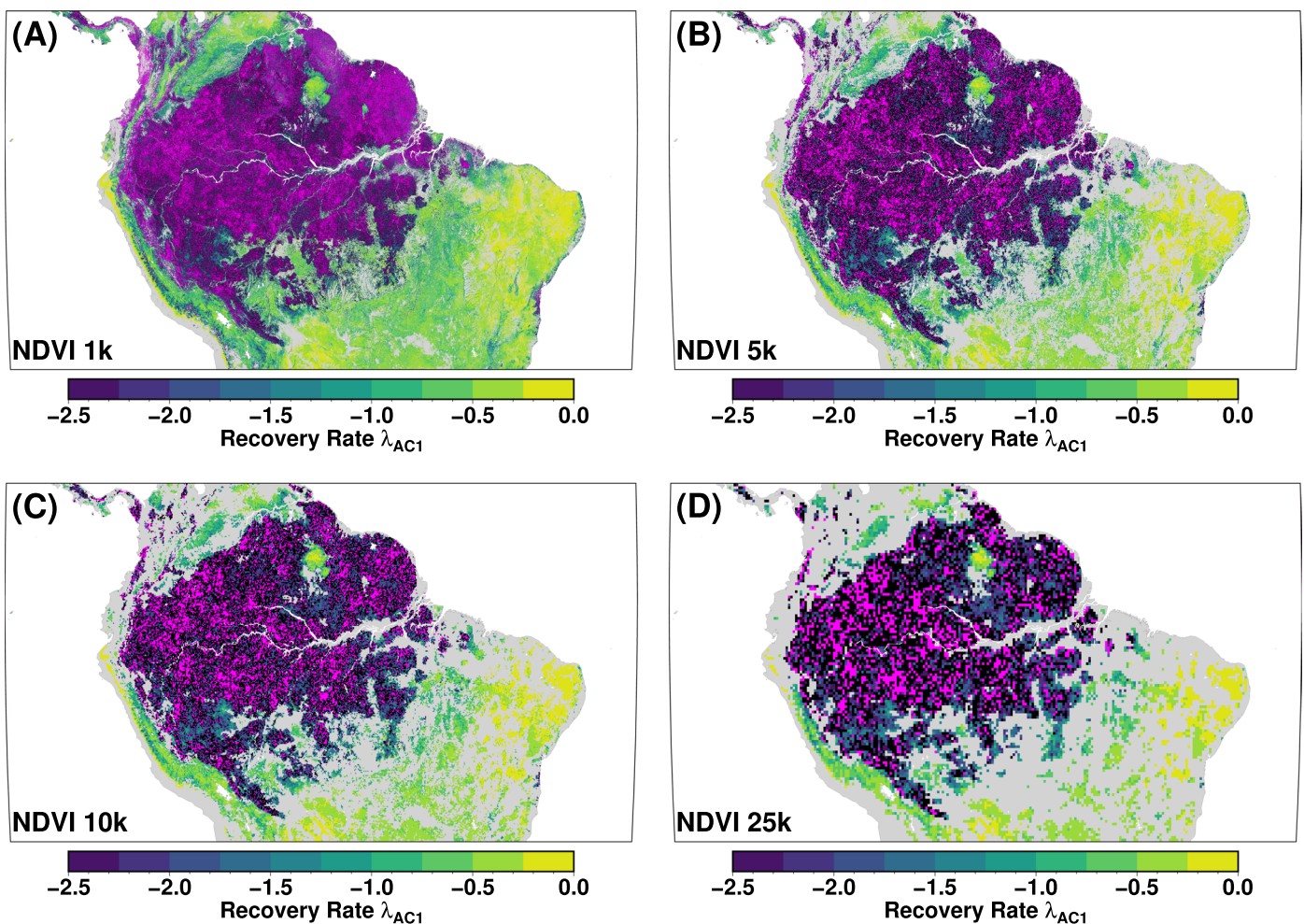

**Extended Data Fig. 7 | Long-term NDVI-based $\lambda_{AC1}$ over the Amazon showing the results of spatial aggregation on data gaps.** (A) 1 km, (B) 5 km, (C) 10 km, (D) 25 km data. Grey areas masked for land-cover (see Methods), magenta areas show undefined $\lambda$ estimates.

# Reporting Summary

## Statistics

For all statistical analyses, confirm that the following items are present in the figure legend, table legend, main text, or Methods section.

| n/a | Confirmed | |
|---|---|---|
| ☐ | ☒ | The exact sample size (*n*) for each experimental group/condition, given as a discrete number and unit of measurement |
| ☒ | ☐ | A statement on whether measurements were taken from distinct samples or whether the same sample was measured repeatedly |
| ☐ | ☒ | The statistical test(s) used AND whether they are one- or two-sided<br>*Only common tests should be described solely by name; describe more complex techniques in the Methods section.* |
| ☐ | ☒ | A description of all covariates tested |
| ☒ | ☐ | A description of any assumptions or corrections, such as tests of normality and adjustment for multiple comparisons |
| ☐ | ☒ | A full description of the statistical parameters including central tendency (e.g. means) or other basic estimates (e.g. regression coefficient) AND variation (e.g. standard deviation) or associated estimates of uncertainty (e.g. confidence intervals) |
| ☒ | ☐ | For null hypothesis testing, the test statistic (e.g. *F*, *t*, *r*) with confidence intervals, effect sizes, degrees of freedom and *P* value noted<br>*Give P values as exact values whenever suitable.* |
| ☒ | ☐ | For Bayesian analysis, information on the choice of priors and Markov chain Monte Carlo settings |
| ☒ | ☐ | For hierarchical and complex designs, identification of the appropriate level for tests and full reporting of outcomes |
| ☐ | ☒ | Estimates of effect sizes (e.g. Cohen's *d*, Pearson's *r*), indicating how they were calculated |

*Our web collection on statistics for biologists contains articles on many of the points above.*

## Software and code

Policy information about availability of computer code

| Data collection | Data assimilation and processing was done using Python [v. 3.9.13] and Google Earth Engine, based on publicly available data. |
|---|---|
| Data analysis | Data analysis was performed using the Python [v. 3.9.13] language. Analysis codes can be found on Zenodo: 10.5281/zenodo.7550255 |

For manuscripts utilizing custom algorithms or software that are central to the research but not yet described in published literature, software must be made available to editors and reviewers. We strongly encourage code deposition in a community repository (e.g. GitHub). See the Nature Portfolio guidelines for submitting code & software for further information.

## Data

Policy information about availability of data

All manuscripts must include a data availability statement. This statement should provide the following information, where applicable:

- Accession codes, unique identifiers, or web links for publicly available datasets
- A description of any restrictions on data availability
- For clinical datasets or third party data, please ensure that the statement adheres to our policy

We use MODIS EVI and NDVI data (products MOD13Q1 and MOD13A2), as well as GPP (MOD17A2) and LAI (MCD15A3H). We further use land-cover data from MODIS (MCD12Q1). The raw data used in this study are all available via Google Earth Engine. Codes to process these data are provided on Zenodo: 10.5281/zenodo.7550255.

# Research involving human participants, their data, or biological material

Policy information about studies with human participants or human data. See also policy information about sex, gender (identity/presentation), and sexual orientation and race, ethnicity and racism.

| | |
|---|---|
| Reporting on sex and gender | N/A |
| Reporting on race, ethnicity, or other socially relevant groupings | N/A |
| Population characteristics | N/A |
| Recruitment | N/A |
| Ethics oversight | N/A |

Note that full information on the approval of the study protocol must also be provided in the manuscript.

# Field-specific reporting

Please select the one below that is the best fit for your research. If you are not sure, read the appropriate sections before making your selection.

☐ Life sciences   ☐ Behavioural & social sciences   ☒ Ecological, evolutionary & environmental sciences

For a reference copy of the document with all sections, see nature.com/documents/nr-reporting-summary-flat.pdf

# Ecological, evolutionary & environmental sciences study design

All studies must disclose on these points even when the disclosure is negative.

| | |
|---|---|
| Study description | In this study, we systematically compare the methods and data used in several recent publications for estimating vegetation resilience at the global scale. We first examine methodological techniques, before applying a chosen optimal method to several different vegetation data sets. We finally use only those regions with reliable resilience estimates to examine changes in vegetation dynamics through time. |
| Research sample | We sampled all vegetated areas which had not been significantly influence by humans (e.g., farms, urban areas) and had not changed land-cover types (e.g., from Forest to Savanna) over the study period. We excluded human-influenced areas to study only natural changes in ecosystems (i.e. those not caused by e.g. agriculture). This analysis was global, and relied on multiple satellite data sets, all of which are publicly available (MOD13Q1, MOD13A2, MOD17A2, MCD15A3H, MCD12Q1).<br><br>In a second step, we further sample 100,000 random locations for more in-depth analysis in order to compare different land cover types. These sample locations are availabe on Zenodo. |
| Sampling strategy | We have two samples. The first is all vegetated areas without human influence. The second is 100,000 samples, chosen by a stratified random sample to ensure an equal number of samples per land cover type (n=10,000 for each of 10 natural land cover types). Code to generate and duplicate our sampling approach is available on Zenodo. |
| Data collection | Data was collected by NASA (MODIS data) and was accessed via Google Earth Engine. We did not perform any further data collection. |
| Timing and spatial scale | We used available MODIS data from Oct 2000 to Oct 2022 to cover complete years. We used both native-resolution data (down to 250 m), as well as resampling our data spatially (1, 5, 10, 25 km) to mimic the data resolutions used in previous research and examine the role of spatial aggregation in resilience estimation. |
| Data exclusions | Data was excluded if there was significant human land cover, as we could no longer look for relationships in natural vegetation in this case. E.g., farms do not follow a natural annual water cycle, but rather respond to human-induced watering changes. |
| Reproducibility | All codes needed to reproduce our results are available in Zenodo. All data is open source. |
| Randomization | Data was divided primarily by land-cover type. This is a necessity when comparing vegetation with different basic functions -- the inherent speed of plant growth varies from place to place and by ecosystem. We used a random sample from each land cover type to ensure that all land cover types are sampled equally, despite covering different amounts of the Earth. |
| Blinding | Blinding was not relevant to our study. |

Did the study involve field work?   ☐ Yes   ☒ No

# Reporting for specific materials, systems and methods

We require information from authors about some types of materials, experimental systems and methods used in many studies. Here, indicate whether each material, system or method listed is relevant to your study. If you are not sure if a list item applies to your research, read the appropriate section before selecting a response.

## Materials & experimental systems

| n/a | Involved in the study |
|-----|----------------------|
| ☒ | ☐ Antibodies |
| ☒ | ☐ Eukaryotic cell lines |
| ☒ | ☐ Palaeontology and archaeology |
| ☒ | ☐ Animals and other organisms |
| ☒ | ☐ Clinical data |
| ☒ | ☐ Dual use research of concern |
| ☒ | ☐ Plants |

## Methods

| n/a | Involved in the study |
|-----|----------------------|
| ☒ | ☐ ChIP-seq |
| ☒ | ☐ Flow cytometry |
| ☒ | ☐ MRI-based neuroimaging |

