## [Peer Review File · Nature Ecology & Evolution]

Peer Review Information

Journal: Nature Ecology & Evolution

Manuscript Title: Reliability of Vegetation Resilience Estimates Depends on Biomass Density

Corresponding author name(s): Taylor Smith

Editorial Notes:

Reviewer Comments & Decisions:

Decision Letter, initial version:

9th May 2023

Dear Dr Smith,

Your manuscript entitled "Reliability of Vegetation Resilience Estimates Depends on Biomass Density" has now been seen by 3 reviewers, whose comments are attached. The reviewers have raised a number of concerns which will need to be addressed before we can offer publication in Nature Ecology & Evolution. We will therefore need to see your responses to the criticisms raised, along with a revised manuscript, before we can reach a final decision regarding publication.

* If you have not done so already please begin to revise your manuscript so that it conforms to our Article format instructions at <http://www.nature.com/natecolevol/info/final-submission>. Refer also to any guidelines provided in this letter.

[REDACTED]

Note: This URL links to your confidential home page and associated information about manuscripts you may have submitted, or that you are reviewing for us. If you wish to forward

2this email to co-authors, please delete the link to your homepage.

Nature Ecology & Evolution is committed to improving transparency in authorship. As part of our efforts in this direction, we are now requesting that all authors identified as 'corresponding author' on published papers create and link their Open Researcher and Contributor Identifier (ORCID) with their account on the Manuscript Tracking System (MTS), prior to acceptance. ORCID helps the scientific community achieve unambiguous attribution of all scholarly contributions. You can create and link your ORCID from the home page of the MTS by clicking on 'Modify my Springer Nature account'. For more information please visit www.springernature.com/orcid.

[REDACTED]

Reviewers' comments:

Reviewer #1 (Remarks to the Author):

Dr. Smith and dr. Boers have evaluated the impact of preprocessing techniques (including spatial aggregation, temporal compositing, time series decomposition, and handling data gaps) on resilience indicators. In addition, they assessed the reliability of resilience indicators by comparing the level of trend agreement between two indicators: the autocorrelation at lag one and variance.

Evaluating the impact of pre-processing techniques and the reliability of resilience indicators is critical. A range of preprocessing methods has been applied in studies focusing on (large-scale) resilience patterns and validation of resilience indicators with in-situ data is challenging.

I have a few questions/suggestions though:

1. My main concern is the set-up of the simulations, which is not entirely clear to me (line 282 – 290). How do you ensure that the simulated time series realistically represent satellite time series? Since you work at global scale, you face a lot of variability in the time series characteristics. For instance, a sine curve has been added to represent seasonality, but the shape of the seasonal pattern is expected to differ amongst land cover types. Is the outcome of the simulation study insensitive to this effect? Did you test seasonal patterns that change over time (e.g. changes in amplitude and timing)? How do

2you define the amplitude of your seasonal pattern? How do you define the level of noise? How many time stamps per year, how many years were simulated? Is both the level and the overall trend in lag-1 autocorrelation and variance that you simulate similar to the level and trends you find in empirical time series?

2. Although a mismatch between the lag-one autocorrelation and variance is an indication of unreliable estimates, the agreement of trends is not necessary a proof that the trends are correct. For instance, - although the authors tried to minimize the effects of anthropogenic impact - it is hard to fully exclude this. It could be worth considering to discuss this.

3. Line 299 and figure 1: You could add the variability around the average AC1. Although STL seems to overfit the data, it's trend seems more consistent compared to the other two methods. I wonder if different STL settings wouldn't resolve the overfitting issue?

4. When evaluating the effect of gaps using simulated time series (e.g. figure S2), you compute the lag1 autocorrelation on a sample of 100 simulated series. Do you use the same time series length as the moving window approach (so 5 years) you applied on the MODIS time series? How many observations do you have when you fit the regression line through the lag-1 autocorrelation of the MODIS time series? Would the time series length and number of samples affect the result?

5. Figure 2 was difficult to read, the labels (and text in general) are very small.

6. Line 191: why do you expect mixed time series to be smoother than pure ones? Isn't this also due to a the reduction of noise, e.g. residual cloud effects, ... that gets averaged out?

7. Line 162: It could perhaps be interesting to add a plot to S11 and S12 with the number of VI's that agree on the trend direction.

Reviewer #2 (Remarks to the Author):

The paper explores the validity of the application of early warning indicators on different, publicly available optical vegetation indices and using different data processing approaches. Based on ecological theory both variance and autocorrelation should correlate and increase as a system approaches a tipping point as a result of decreased resilience. The most important finding of the paper is that for many ecosystems worldwide this criteria is not fulfilled, particularly in forested ecosystems, probably as a result of the challenges in generating high quality satellite data that is free from cloud and saturation artefacts, or as a result of short growing seasons. This finding is important because it potentially calls into question the results many of the recently published estimates of resilience based on similar and associated methods. For the locations where the two criteria are fulfilled the authors provide maps of locations of resilience increase and decreases.

3The analysis is well done and the conclusions in general I think the recommendations for processing the data are solid and well thought out. I also support the overall objectives and main messages of the paper. It is probably too easy to download and process optical vegetation indices, and to then analyse them under the general framework of resilience without consideration of the quality of the data and/or the appropriateness of the metric used. These authors have here, and in other recent papers, made important steps in advancing the methods related to spatial metrics of resilience worldwide.

Nevertheless, I do think the paper could be improved in various ways. I outline these suggestions here:

1) There should be more of a theoretical discussion in the introduction about critical slowing down, autocorrelation, variance, and the recovery rate and how this linked to the 'restoring rate' derived from the equations in your paper. In particular the difference between the restoring rate and the recovery rate is not obvious from the introduction alone, nor even when introduced again on line 119 (without going all the way to the end of the methods). The authors could do more to help the reader along with these theoretical concepts. In my opinion this could be solved with a conceptual figure and an associated paragraph in the first part of the introduction.

2) The main point of the paper is to use critical slow down as a proxy for ecological resilience, which results in both increases in both autocorrelation and variance as predicted in theory. The idea behind the paper is that if you don't have both these characteristics covarying in the same direction then the critical slowdown estimates are unreliable and should be discounted.

But I think it is important to note in the paper that there are other ways critical transitions can happen where the autocorrelation and variance don't covary. For example, this paper:

<https://www.nature.com/articles/nature11655> (Wang et al. 2012) shows increased variance can occur in conjunction with decreased autocorrelation in their model as a critical transition is approached as indicated by flickering (although empirically their analysis is flawed by the data processing they use, as demonstrated by the subsequent comment attached to the paper). Contrarily this paper by Dakos et al. 2013 (<https://link.springer.com/article/10.1007/s12080-013-0186-4>) shows flickering can resemble critical slowdown in variance and autocorrelation despite quite different dynamics. Finally, one can have critical slowdown in systems that aren't experiencing a critical transitions (Kefi et al. 2013, <https://onlinelibrary.wiley.com/doi/abs/10.1111/j.1600-0706.2012.20838.x>). Systems might not need to show any such EWS if they experience a large and very sudden shock.

Thus, I think the paper needs to highlight that the criteria they set based on rising variance and increased autocorrelation could indeed reflect critical slowdown, but there are other potential system dynamics that are equally important to acknowledge that it could or could not reflect. Dakos et al. 2013 (see link above) describe critical slowdown as "the exception rather than the rule", and since most successful detections of critical slowdown in relation to critical transitions have been conducted under controlled environments, we have to appreciate that the methods they are presenting are just one potential interpretation of the overall system dynamics. Finally, of the systems where the criteria of rising variance and autocorrelation are fulfilled, are these vegetation types actually known to follow the behaviour assumed by the model and demonstrate alternative stable states? (e.g. can you provide an assessment of the potential feedbacks/ mechanisms for the systems that their analysis identifies as

4reliably demonstrating critical slowing down?).

3) The authors do a lot to identify best practices for processing the optical vegetation indices for such analysis, and go on to produce maps where rising and falling resilience can be identified based on the assumptions of critical slowdown. There are now many papers published with similar analyses and findings. Apart from removing certain regions because of inconsistent/ undefined resilience metrics, there is very little done to actually compare the spatial patterns of this study and to identify the added value of this study compared to previous ones in terms of the overall understanding of spatial patterns (i.e. what regions are consistently identified as being identified as gaining/ lacking resilience in these papers; what are the key differences in spatial patterns and can these differences be explained by the improved methods used in your model). Similarly, only very little attention is provided to explaining the potential processes that potentially govern the key patterns.

Reviewer #3 (Remarks to the Author):

Overall comments

This paper deals with the assessment of 'resilience' of vegetation types through the 'Critical Slowing Down' (CSD) approach, using MODIS satellite data series. CSD became popular for anticipating bifurcation-induced transition (tipping), from the expectation that simultaneous increases in series lag-one autocorrelation (AC1) and variance are fingerprints of CSD and therefore could be early warning signals of catastrophic shifts. Note that if CSD is well-defined, 'resilience' is not, so directly stating that CSD indicator is measuring 'resilience' as it is done throughout the introduction section is far-fetched (the Methods part is properly phrased) Anyway, accurately estimating AC1 and variance in satellite data series is indeed critical for detecting CSD

This paper investigates several methodological difficulties that may cripple the approach and determine irrelevant AC1/Variance estimates and/or false interpretations, namely:

- Methods for data preprocessing (detrending and deseasoning), since series need to be stationary for CSD indicator assessment;
- Comparing the performance of the five MODIS vegetation indices;
- Assessing the effects of gaps in series (eg due to cloudiness, winter snow, ...) and of spatial aggregation of MODIS pixels (from sensor-native resolution).

However, it seems to me that the main contribution of this paper to the CSD detection topic is providing close form results (Eqs 4 to 8 in Methods) for assessing both lag-1 correlation and variance of the series, and to draw attention to the necessary condition that AC1 and variance-based assessments must agree to ensure the plausibility of the results.

However, I didn't take this as the main message of the paper in the introduction section, but rather later in the paper. Other readers may also miss it, as the above-mentioned technical points are highlighted and catch the readers' attention.

In addition, I feel that the authors should clarify whether results leading to Eqs 4 to 8 are original or not. In the first case it is an achievement, while in the second references are needed (see below)

The main empirical results showed problematic estimation of CSD-based indicators for dense forest vegetation of both boreal and tropical areas. On the other hand, acceptable scatter around the expected one to one relationship between AC1- and variance-based estimates was observed in open

5vegetation (mainly in drylands) for which spatially consistent estimations of CSD indicators are obtained.

Regarding the methods points, the authors concluded that (i) not all methods of removing trend and seasonality work equally well, especially in the presence of gaps (and they propose a pre-processing scheme); (ii) spatial aggregation (ie decreasing resolution) generally helps stabilizing estimates. Overall, the paper provides a solid technical contribution to the CSD detection topic along with a global scale illustration. It deserves improvements to foster the interest from a broad leadership.

Specific points

L113 It's useful, but it's quite late to explain that `

L117. Please provide here a reference for Orstein Uhlenbeck processes

L121 'indicating that CSD is only appropriate to quantify resilience for certain vegetation types.' This is suggesting that the relationship between CSD detection and resilience quantification is far from being unequivocal. This important point is unfortunately kept implicit in the paper.

L124 It is indeed related to biomass but not as 'closely' as suggested.

L143-145 'However, our results show that this problem for tropical forests is not only present for the NDVI, but also for kNDVI, EVI, and LAI, as well as GPP to lesser degree'. So the problem is intrinsic to dense canopy.

Fig4. Why is Sahara gray and not central Australia?

L175 '..., (iii) differences in baseline values of CSD-based resilience estimates, as well as in their reliability'. What is meant by 'baseline values' is unclear for me (see also below, eg L198)

L180-181. 'Our proposed methodology not only performs similarly to a 'perfect' pre-processing setup, ...'. A less assertive tone would be welcome. Indeed, it behaves 'perfectly' with respect to a particular type of 'paradigmatic' simulated data.

L194. 'Confident calculation'. Which criterion is used to assess 'confidence'?

L209-2011. Here you propose an interpretation for problems in boreal forests – fine! Could you propose some explanation why close canopy tropical forest also raises problems? I agree that some 'issues' (L216 – note that no reference is provided) have been frequently evoked in literature but it could be good to recall briefly what issues seem important for the CSD-estimates topic.

L236-237. Some other areas seem to show increased 'resilience'. Why not mentioning and discussing them?

L246-247. It was stated above that all MODIS indices poorly perform wrt dense vegetation (moreover also recalled in the next sentence). Why here advocating GPP so strongly?

L273. What was the script language used? Just specify to let the reader know without having to turn to Zenodo

L322 and subsequent. There are some fundamental (and interesting) considerations provided here. For already established results some references should be provided (eg for Ornstein-Uhlenbeck processes) this is an ecology and evolution journal). Original results (if any) would deserve to be underlined (wrt eqs (4) to (8))

*****END*****

Author Rebuttal to Initial commentsReply to Reviewers - Nature E&E

Reviewer #1 (Remarks to the Author):

Dr. Smith and Dr. Boers have evaluated the impact of preprocessing techniques (including spatial aggregation, temporal compositing, time series decomposition, and handling data gaps) on resilience indicators. In addition, they assessed the reliability of resilience indicators by comparing the level of trend agreement between two indicators: the autocorrelation at lag one and variance.

Evaluating the impact of pre-processing techniques and the reliability of resilience indicators is critical. A range of preprocessing methods has been applied in studies focusing on (large-scale) resilience patterns and validation of resilience indicators with in-situ data is challenging.

Thank you for taking the time to review our manuscript and your valuable suggestions. We will respond to each of your questions in-line in the following.

I have a few questions/suggestions though:

1. My main concern is the set-up of the simulations, which is not entirely clear to me (line 282 – 290). How do you ensure that the simulated time series realistically represent satellite time series? Since you work at a global scale, you face a lot of variability in the time series characteristics. For instance, a sine curve has been added to represent seasonality, but the shape of the seasonal pattern is expected to differ amongst land cover types. Is the outcome of the simulation study insensitive to this effect?

Thank you for this comment. We agree that, given that we investigate global data, the characteristics of the time series will vary greatly for different regions. However, with the simulation study we wanted to first highlight difficulties inherent in deseasoning any data; starting from a simple model allows us to identify problems that may arise even in the most basic case. For example, the widely used approach for deseasoning, based on subtracting long-term means of each day or month, has severe problems even in the simplest setting of a regular sine-like seasonality. We have emphasized this in the revised methods section.

In our revision, we have added several additional simulations to address your concerns, and more deeply explored issues around deseasoning. We will describe these different additions in-line below with regards to your further specific questions about the simulation setup.

Is both the level and the overall trend in lag-1 autocorrelation and variance that you simulate similar to the level and trends you find in empirical time series?

Our model is set up to simulate time series that approach a critical transition. In line with the theory described in our methods, the lag-one autocorrelation increases gradually from values close to zero toward a value of one, which marks the theoretical critical transition point. Hence, the levels of lag-one autocorrelation cover the entire possible range and the trend is set by how

fast we change the critical control parameter. We think that it therefore does not make sense to match the levels and trends between our synthetic example (where we enforce a transition) and the empirical time series (which rarely have such an abrupt state transition).

Did you test seasonal patterns that change over time (e.g. changes in amplitude and timing)? How do you define the amplitude of your seasonal pattern? How do you define the level of noise? How many time stamps per year, how many years were simulated?

Thank you for these important questions. In our original submission we aimed to set up a minimal, stylized toy model that includes (only) a nonlinear trend, a stationary seasonality, and white noise, in order to test the performance of different de-seasoning techniques used in recent papers in a comparably simple setup. The rationale behind this was to assure that one can easily trace back issues that different methods might have, which would be difficult with realistic time series. By using a simple model, we can isolate that one impact of de-seasoning – namely how stationary the resulting de-seasoned time series are, and whether we detect spurious signals or not.

Changing the seasonal amplitude does not make a difference in any of the tested preprocessing methods, as they all conform well to any seasonal amplitude. The sigma of the noise was set to 0.2 based on our previous experience with these kinds of simple models (see Smith et al., 2023 ESD). In essence, both parameters were chosen to qualitatively match highly seasonal vegetation patterns. The different methods also work well with higher (or, of course, lower) sigma. We simulated 365 values per year, over 31 years, basically corresponding to the longest available satellite records.

In our original submission, however, we did not include varying amplitudes or timings of the seasonal cycle. To address this concern, we have included in the MS an additional, more complex, version of the model which uses variable seasonal timing and amplitudes. This can be seen below in Figure 1. Thank you for this suggestion!

Figure 1 - Modified Figure 1 of the main text, showing on the right a variable seasonal model. Both timing and amplitude of the seasonal cycle are randomized year-on-year. The time before the abrupt transition has been lengthened to make differences between the methods easier to see. Rolling AC1 windows (bottom panels) are now plotted at the center of the moving window.

In addition, we did robustness checks on (1) changing the seasonality to have two peaks, (2) increasing the sigma of the noise, and (3) modifying the STL parameters. These can be seen below in Figures 2 and 3, and in Supplementary Figures 18 and 19. As we fit a third-order harmonic to deseason our data, multiple seasonal peaks are well-captured by our method. Similarly, higher noise remains in the residual, but if it is uncorrelated it does not strongly influence the trends in AC1 through time. Using alternative STL settings also does not change the results significantly.

Figure 2 - Alternative Model Setups. In each case, the differences between deseasoning methods are similar. This has also been added to the Supplement as Figure S18.

The hardest type of signal to remove is one with changing amplitude/timing over the course of the simulation; this is true for all deseasoning methods considered. If those amplitude changes are randomly distributed, they will increase AC1 values but not bias the trends in AC1. If the seasonal amplitude increases monotonically through time, however, there will be more seasonality left in the residual for later times (since we fit a single harmonic to the whole time series), which will induce an increase in AC1.

It strongly depends on the specific setting whether that is a good or a bad thing – on one hand, we want to end up with a stationary time series, but on the other hand, such a strong change in seasonality can also be seen as a change in the system and hence a marker of a changing (eco)system dynamics. In particular, in cases of varying seasonal amplitudes, whether one would want to remove it entirely or not depends critically on the question whether the changing seasonality is expressed only in the response of the vegetation system, or rather due to changing seasonality in the weather / climate forcing. In the first case, a deseasoning method that would remove the entire, variable seasonality, might mask signals that could be indicative of resilience loss, whereas in the second case, one would indeed want to remove the varying seasonality altogether. We have added this consideration in our revised manuscript.

With regards to deseasoning methods, STL tends towards being a more conservative approach, mainly through lowered overall AC1 values (Figure 1 of this Reply). On this basis, we conclude that it depends on the specific problem whether the harmonic fitting or the STL should be preferred and in our revised manuscript, we recommend that comparisons between results based on harmonic fitting and on STL should be made before drawing conclusions.

Figure 3 - Alternative model setups, illustrating cases where STL is more conservative than harmonic fitting. Left side: Varying seasonal timing, right side: varying seasonal amplitude.

After performing these additional model simulations, we have decided to include an additional analysis in our manuscript – namely, reproducing our results using data de-seasoned via STL. In our simulation tests, we found that STL is a more conservative way to measure changes in resilience through time – to the point of being too conservative compared to the ‘perfect’ deseasoning case in some instances.

For the global satellite vegetation data, we found that while in some cases STL performs very well, it is highly sensitive to data gaps. We used a gap-aware implementation of STL (<https://github.com/mortvest/hastl>), but this is only designed to handle short data gaps (the original FORTRAN implementation of STL commonly used does not support missing data at all). For example, when using a very noisy time series (cf. Figure 4 of this Reply), or very gappy time series, STL cannot derive adequate spline fits and does not return a residual. This a clear caveat of the STL method when applied to observational data. We emphasize that we strongly advise against any gap-filling or interpolation of the underlying data when investigating CSD-based resilience indicators, as this can dramatically bias the computed trends in CSD indicators (see e.g. [Ben-Yami et al., in revision (<https://arxiv.org/abs/2303.06448>)] for a full discussion of this). Although it may be considered a gold standard for deseasoning and detrending, the problems that STL has with data gaps motivates the alternative deseasoning and detrending routine that we propose in our manuscript, based on smoothing and fitting of higher-order harmonics. As our results show, our method performs similarly to STL in cases where the latter is applicable, but is less sensitive to gaps, without the need for any interpolation or gap filling.

Figure 4 - Very noisy and gappy time series taken from the Amazon. STL does not yield a defined residual.

In practice, when we run our analysis globally, we find that the spatial patterns obtained using STL for data preprocessing are qualitatively very similar to what we show using the harmonic deseasoning approach (Figure 5).

Figure 5 - Reproductions of Figure 3A (left) and Figure 4 (right) of the main text, when using STL to deseason NDVI data. These are also included in the updated Supplement as Figures S11 and S15.

As expected, the overall lambda values are lower (left panel), as STL tends to overfit the time series (see Figure 1G of this Reply). Furthermore, there are far fewer inconsistent trends (right panel) than with the harmonic fitting approach – these have mostly been captured by the ‘Undefined’ class, indicating that STL did not return a useful residual. The number of ‘Undefined’

residual time series has increased, but the spatial pattern of recovery rates and trends therein is very similar to that obtained via the harmonic deseasoning. This indicates that while STL performs better in some regions, it does a worse job of handling gappy or otherwise very noisy data than the harmonic fitting approach, when considering all time series globally. It could also be thought of as a more conservative approach, as it will only yield useful residuals for very well-behaved time series (hence the strong decrease in mixed/inconsistent recovery rate trends).

We have included figures showing this analysis for all vegetation indices in the Supplement, and have added additional discussion to the manuscript surrounding the use of STL, limitations of the different deseasoning methods (especially with regards to data gaps), and implications for choosing the right deseasoning procedure. These results do not change our main conclusions, but rather give an additional robustness check on our methodology and confirm that optical data is poorly suited for vegetation resilience analysis in many regions.

2. Although a mismatch between the lag-one autocorrelation and variance is an indication of unreliable estimates, the agreement of trends is not necessarily proof that the trends are correct. For instance, - although the authors tried to minimize the effects of anthropogenic impact – it is hard to fully exclude this. It could be worth considering to discuss this.

Thank you for this point – it is indeed difficult to entirely remove anthropogenic signals, for example in forests with dense canopy where human activity might go unnoticed by satellites. We endeavored to be very conservative with our land-cover masking (and indeed much more conservative than previous studies), but we agree with the referee that there remains the possibility of human influence in 'natural' regions of the world. We have added further caveats to our Discussion section with regards to this.

3. Line 299 and figure 1: You could add the variability around the average AC1. Although STL seems to overfit the data, its trend seems more consistent compared to the other two methods. I wonder if different STL settings wouldn't resolve the overfitting issue?

We have included here a version of Figure 1 showing (1) the variation around the mean (cf. Figure 1 of this Reply), and (2) different STL settings (cf. Figure 2 of this Reply). STL performs well across all settings we tested, but still overfits to the underlying data when compared to a harmonic fit (the lag-one autocorrelation values for STL are systematically lower than in the case of 'perfect deseason / detrend'). This is because of the nested local nature of the spline fits (vs a global harmonic fit), which tend towards fitting to fine-scale variability. We tested several moving windows, spline fit orders, and filter lengths, with basically identical results for our simple toy model.

4. When evaluating the effect of gaps using simulated time series (e.g. figure S2), you compute the lag1 autocorrelation on a sample of 100 simulated series. Do you use the same time series length as the moving window approach (so 5 years) you applied on the MODIS time series?

For Figure S2, we use the very same model setup as in Figure 1 of the main MS – namely a sinusoidal model with seasonality removed via harmonic fitting. We use the same 5-year rolling window as with the MODIS data, and the time series are of a similar length (~25 years). We have clarified this in our revised manuscript.

How many observations do you have when you fit the regression line through the lag-1 autocorrelation of the MODIS time series? Would the time series length and number of samples affect the result?

The length of the NDVI/EVI/kNDVI time series is roughly 500 points (16 day, 2000-2022), see Supplemental Figure S12 for a map of the exact numbers of data points used for each location. GPP and LAI have more data points, as they are used at 8-day and 4-day time steps, respectively. The length/number of samples of the MODIS data would of course impact the result of the trend detection – if there were too few values (e.g., very short or very gappy series), we would not get a good harmonic fit and hence a poorly measured AC1 and AC1 trend. This is to some degree expressed in our global reliability maps (Figure 3 of the main MS), where regions with fewer measurements are less likely to have robust signals (e.g., cloudy Amazon, high-latitude regions impacted by snow cover). This is one of the key points we wanted to make with this paper – not all regions are suitable for the estimation of vegetation resilience from optical satellite data.

5. Figure 2 was difficult to read, the labels (and text in general) are very small.

We have increased the label sizes here to make this clearer.

6. Line 191: why do you expect mixed time series to be smoother than pure ones? Isn't this also due to a reduction of noise, e.g. residual cloud effects, ... that gets averaged out?

Yes, this is exactly what we mean here – when you take a local average (e.g., compress 250m pixels into a 10km mean), you remove some noise. This results in a smoother signal. We have made this clearer in that line.

7. Line 162: It could perhaps be interesting to add a plot to S11 and S12 with the number of VI's that agree on the trend direction.

Figure 6 - Number of instruments agreeing on trend directions.

Thank you very much for this suggestion! We fully agree and have included these plots in the Supplement (Figure S16), and here as well for reference.

Reviewer #2 (Remarks to the Author):

The paper explores the validity of the application of early warning indicators on different, publicly available optical vegetation indices and using different data processing approaches. Based on ecological theory both variance and autocorrelation should correlate and increase as a system approaches a tipping point as a result of decreased resilience. The most important finding of the paper is that for many ecosystems worldwide this criteria is not fulfilled, particularly in forested ecosystems, probably as a result of the challenges in generating high quality satellite data that is free from cloud and saturation artefacts, or as a result of short growing seasons. This finding is important because it potentially calls into question the results of many of the recently published estimates of resilience based on similar and associated methods. For the locations where the two criteria are fulfilled, the authors provide maps of locations of resilience increase and decreases.

Thank you for taking the time to review our paper, and for this accurate summary!

The analysis is well done and the conclusions in general I think the recommendations for processing the data are solid and well thought out. I also support the overall objectives and main messages of the paper. It is probably too easy to download and process optical vegetation indices, and to then analyze them under the general framework of resilience without consideration of the quality of the data and/or the appropriateness of the metric used. These authors have here, and in other recent papers, made important steps in advancing the methods related to spatial metrics of resilience worldwide.

Thank you for the overall positive evaluation of our manuscript! We are glad that the main message was well conveyed – there are important points to consider before applying these methods (as has been done quite often in the past years), and a detailed treatment of the reliability of these resilience methods was lacking. We will answer your comments and questions in-line in the following.

Nevertheless, I do think the paper could be improved in various ways. I outline these suggestions here:

1) There should be more of a theoretical discussion in the introduction about critical slowing down, autocorrelation, variance, and the recovery rate and how this is linked to the 'restoring rate' derived from the equations in your paper. In particular the difference between the restoring rate and the recovery rate is not obvious from the introduction alone, nor even when introduced again on line 119 (without going all the way to the end of the methods). The authors could do more to help the reader along with these theoretical concepts. In my opinion this could be solved with a conceptual figure and an associated paragraph in the first part of the introduction.

We did not include such a conceptual figure and had kept the theoretical description relatively short in our initial manuscript because the theory of CSD has been discussed thoroughly in several recent papers [Scheffer et al., Nature 2009; Boers & Rypdal PNAS 2021, Boers, Nature Climate Change 2021; Smith et al., Nature Climate Change 2022, Boers et al., Env. Res. Let.

2022]. We would prefer not to show a sketch as suggested in the main text because of space limitations, as well as the ubiquity of such theoretical sketches in other recent papers. However, we agree that we should take some more time going through the theoretical background of CSD in the introduction, and have added additional text detailing how we get from theory to practice within this framework; we have also added several references to recent papers explaining the CSD concept in detail there.

2) The main point of the paper is to use critical slow down as a proxy for ecological resilience, which results in both increases in both autocorrelation and variance as predicted in theory. The idea behind the paper is that if you don't have both these characteristics covarying in the same direction then the critical slowdown estimates are unreliable and should be discounted. But I think it is important to note in the paper that there are other ways critical transitions can happen where the autocorrelation and variance don't covary. For example, this paper: <https://www.nature.com/articles/nature11655> (Wang et al. 2012) shows increased variance can occur in conjunction with decreased autocorrelation in their model as a critical transition is approached as indicated by flickering (although empirically their analysis is flawed by the data processing they use, as demonstrated by the subsequent comment attached to the paper). Contrarily this paper by Dakos et al. 2013 (<https://link.springer.com/article/10.1007/s12080-013-0186-4>) shows flickering can resemble critical slowdown in variance and autocorrelation despite quite different dynamics. Finally, one can have critical slowdown in systems that aren't experiencing a critical transitions (Kefi et al. 2013, <https://onlinelibrary.wiley.com/doi/abs/10.1111/j.1600-0706.2012.20838.x>). Systems might not need to show any such EWS if they experience a large and very sudden shock.

Thus, I think the paper needs to highlight that the criteria they set based on rising variance and increased autocorrelation could indeed reflect critical slowdown, but there are other potential system dynamics that are equally important to acknowledge that it could or could not reflect. Dakos et al. 2013 (see link above) describe critical slowdown as as "the exception rather than the rule", and since most successful detections of critical slowdown in relation to critical transitions have been conducted under controlled environments, we have to appreciate that the methods they are presenting are just one potential interpretation of the overall system dynamics.

Thank you for this point – this is very well taken and we fully agree that 1) there may be other reasons for AC1 and Variance to increase, without an approaching critical transition or 2) transitions might occur due to other reasons than a bifurcation in the underlying system dynamics, in which case CSD would not yield a warning. We set out in this paper to address the 'most common' approach, which has been used in several recent and visible publications, of looking mainly at the AC1 and variance, based on the framework of CSD. This is, however, only one potential way that a change in resilience can be explored, and we do not spend much time discussing other potential interpretations of the measured changes in the system dynamics. Our focus here was rather on identifying pitfalls and best practices in calculating and assessing the classic CSD metrics (ie, AC1 and variance). However, we have added additional text to our introduction emphasizing potential shortcomings of defining resilience in terms of CSD. We would like to omit the Wang et al paper due to the mentioned caveat regarding their data

processing but have added the paper by V. Dakos (2013) and S. Kefi (2013), thank you for these suggestions! We would also like to note in this context that we have recently shown that empirical recovery rates (as a direct resilience measure) match surprisingly well with AC1- and variance-based estimates of the recovery rate (Smith, Traxl, Boers, Nature Climate Change 2022).

Finally, of the systems where the criteria of rising variance and autocorrelation are fulfilled, are these vegetation types actually known to follow the behavior assumed by the model and demonstrate alternative stable states? (e.g. can you provide an assessment of the potential feedbacks/ mechanisms for the systems that their analysis identifies as reliably demonstrating critical slowing down?).

This is another key point. Some systems have indeed been suggested to have multiple alternative stable states (e.g., savannah vs rainforest in the Amazon region). Others are not so well researched, and don't have clear stable alternatives. It should be emphasized, however, that for the CSD framework to apply, and for measuring resilience based in this framework, bi- or multistability is not needed. As shown in (Smith et al., Nature Climate Change 2022) at a global scale using vegetation optical depth data, the CSD framework can in general be used to estimate the recovery rate of a given vegetation system to external, large perturbations. Assuming that this recovery rate yields a suitable definition of resilience, this provides a strong argument that this CSD framework is indeed suitable across vegetation types and ecosystems. Moreover, our tests in the present paper regarding the consistency between the variance- and AC1-based estimates of the recovery rate show that - for the optical MODIS vegetation indices, the suitability of CSD-based resilience estimates varies greatly by land cover type and biomass density. In this sense, the main research question of our paper addresses this question, independently of the presence of true multistability in the underlying dynamics.

3) The authors do a lot to identify best practices for processing the optical vegetation indices for such analysis, and go on to produce maps where rising and falling resilience can be identified based on the assumptions of critical slowdown. There are now many papers published with similar analyses and findings. Apart from removing certain regions because of inconsistent/ undefined resilience metrics, there is very little done to actually compare the spatial patterns of this study and to identify the added value of this study compared to previous ones in terms of the overall understanding of spatial patterns (i.e. what regions are consistently identified as being identified as gaining/ lacking resilience in these papers; what are the key differences in spatial patterns and can these differences be explained by the improved methods used in your model). Similarly, only very little attention is provided to explaining the potential processes that potentially govern the key patterns.

Thank you for this point. The main goal of our work is to identify potential pitfalls in previous approaches to quantify resilience and its changes from satellite data, and to provide a means of assessing the reliability of these commonly used resilience indicators. As you mention, trends in resilience have been published by several groups using various vegetation data sets, time periods, and processing methods; finding which one is 'best' and why is rather difficult without some sort of ground truth data. Hence, we did not want to focus too strongly on the trends we

found, but rather include them as a logical final step of the analysis – given the drawbacks we found in the methods, there are only some regions where trends should be assessed. Indeed, for many of the key regions discussed in other works (e.g., the Amazon), we find that optical vegetation data is unreliable. Hence, a very important part of Fig. 4 is to show where we conclude that one *cannot* robustly infer a resilience trend. While our data supports the broad conclusions of previous work (e.g., a slight global-scale tendency towards resilience loss), we would hesitate to compare in a more detailed way the derived spatial patterns based on different time periods, spatial resolutions, vegetation data, and processing methods. We do fully agree, however, that a thorough comparison between the different results published in several recent papers, together with an attempt to attribute inferred resilience changes, is needed. This would, however, in our opinion be the subject of a thorough review paper, for fairness also involving the authors from the other mentioned studies.Reviewer #3 (Remarks to the Author):

This paper deals with the assessment of 'resilience' of vegetation types through the 'Critical Slowing Down' (CSD) approach, using MODIS satellite data series. CSD became popular for anticipating bifurcation-induced transition (tipping), from the expectation that simultaneous increases in series lag-one autocorrelation (AC1) and variance are fingerprints of CSD and therefore could be early warning signals of catastrophic shifts. Note that if CSD is well-defined, 'resilience' is not, so directly stating that CSD indicator is measuring 'resilience' as it is done throughout the introduction section is far-fetched (the Methods part is properly phrased) Anyway, accurately estimating AC1 and variance in satellite data series is indeed critical for detecting CSD.

Thank you for taking the time to review the paper. We agree that CSD can only be used as a proxy for resilience and have clarified the steps taken to get from CSD to resilience in our introduction.

This paper investigates several methodological difficulties that may cripple the approach and determine irrelevant AC1/Variance estimates and/or false interpretations, namely:

- Methods for data preprocessing (detrending and deseasoning), since series need to be stationary for CSD indicator assessment;
- Comparing the performance of the five MODIS vegetation indices;
- Assessing the effects of gaps in series (e.g. due to cloudiness, winter snow, ...) and of spatial aggregation of MODIS pixels (from sensor-native resolution).

However, it seems to me that the main contribution of this paper to the CSD detection topic is providing close form results (Eqs 4 to 8 in Methods) for assessing both lag-1 correlation and variance of the series, and to draw attention to the necessary condition that AC1 and variance-based assessments must agree to ensure the plausibility of the results. However, I didn't take this as the main message of the paper in the introduction section, but rather later in the paper. Other readers may also miss it, as the above-mentioned technical points are highlighted and catch the readers' attention. In addition, I feel that the authors should clarify whether results leading to Eqs 4 to 8 are original or not. In the first case it is an achievement, while in the second references are needed (see below).

Thank you for this comment. While we indeed wanted to highlight that the restoring rate λ can be derived from both variance and AC1, we do not feel that this is the main contribution of the paper. Indeed, we have shown a similar derivation in our previous papers (Smith et al., 2022 Nature Climate Change; Smith et al. 2023, Earth System Dynamics); here we wanted to focus on the implications of the calculations of λ which we feel have not been well-addressed in previous work. Hence, we feel that the main contribution is indeed as you found in the introduction – the caveats, best practices, and implications of applying CSD metrics to satellite vegetation data. It should be emphasized, however, that to the best of our knowledge, we are the first to use the theory-motivated requirement of consistency between recovery rates estimated from variance and autocorrelation, respectively, in the present manuscript. This puts us in the unique position to be able to objectively judge on the reliability of CSD-based resilience

estimates for different land cover and vegetation types in practice. We have further detailed our changes to the manuscript to address and clarify this in your specific comments below.

The main empirical results showed problematic estimation of CSD-based indicators for dense forest vegetation of both boreal and tropical areas. On the other hand, acceptable scatter around the expected one to one relationship between AC1- and variance-based estimates was observed in open vegetation (mainly in drylands) for which spatially consistent estimations of CSD indicators are obtained. Regarding the methods points, the authors concluded that (i) not all methods of removing trend and seasonality work equally well, especially in the presence of gaps (and they propose a pre-processing scheme); (ii) spatial aggregation (ie decreasing resolution) generally helps stabilizing estimates. Overall, the paper provides a solid technical contribution to the CSD detection topic along with a global scale illustration. It deserves improvements to foster the interest from a broad readership.

Thank you again for taking the time to review the paper. We will respond to your specific comments in-line.

Specific points

L113 It's useful, but it's quite late to explain that '

We have modified our introduction to make this point earlier, but have left this text here for emphasis alongside our analysis.

L117. Please provide here a reference for Orstein Uhlenbeck processes

Thank you, we have added a reference to the recent review (Boers, Ghil, Stocker, ERL 2022) where this is explained in detail.

L121 'indicating that CSD is only appropriate to quantify resilience for certain vegetation types.' This is suggesting that the relationship between CSD detection and resilience quantification is far from being unequivocal. This important point is unfortunately kept implicit in the paper.

Thank you for this comment. We agree that this is important and feel that this point is actually made in the paper (Discussion, from line 243), but is rather framed around CSD being inappropriate for certain vegetation types. We showed previously (Smith et al., 2022 NCC) that AC1 and Variance-based lambda estimates do indeed follow the theory that they are related to the recovery rate after abrupt transitions, across a range of land-cover types. Hence, we do not think that CSD does not work for certain vegetation types in general, but rather that optical vegetation data does not properly capture the system dynamics in a way that CSD indicators make sense. If we had a better view of the system (for example, using direct biomass estimates, vegetation species mix changes, etc), we could use CSD indicators on that data. However, with optical satellite data we are limited to only those systems whose dynamics are well-captured by NDVI/EVI/etc. Nevertheless, the additional point that CSD is only a proxy to measure resilience via the relationship between CSD indicators to the recovery rate, was indeed not sufficiently clear in our original submission. We have clarified this point in the revised Introduction.

L124 It is indeed related to biomass but not as 'closely' as suggested.

'Closely' in this context is a hard word to define. We found a quite strongly linear relationship between biomass and lambda/lambda correlation (see Figure 7 of this Reply); we feel that this rather strong relationship merits the use of the word closely. We have added this Panel to the SI for completeness (Supplementary Figure S5).

Figure 7 - Reproduction of Figure 2 (bottom panel) without a regularly spaced x-axis. Emphasizes the linear relationship between biomass and correlation between resilience estimates.

L143-145 'However, our results show that this problem for tropical forests is not only present for the NDVI, but also for kNDVI, EVI, and LAI, as well as GPP to a lesser degree'. So the problem is intrinsic to dense canopy.

Yes – this seems to be the case. We proxy this here by Biomass (e.g. Figure 2), but canopy closure could be another strong explanation for this. In essence, optical satellite data does not do a good job of capturing the dynamics of denser vegetation systems. We have added the possibility of other explanatory variables here.

Fig4. Why is Sahara gray and not central Australia?

We followed here the MODIS IGBP land-cover classification, which lists Australia as 'closed shrublands' and 'grasslands'. We did not do any other NDVI thresholding – hence there remains data in Australia.

L175 '...', (iii) differences in baseline values of CSD-based resilience estimates, as well as in their reliability'. What is meant by 'baseline values' is unclear for me (see also below, eg L198)

We refer here to the intrinsic speed of different vegetation systems. Grass grows much faster than forests, for example, and hence has more rapid dynamics. The AC1 of two different types of vegetation hence reflects not only climate, but also intrinsic (baseline) differences in plant physiology. We have clarified this here.

L180-181. 'Our proposed methodology not only performs similarly to a 'perfect' pre-processing setup, ...'. A less assertive tone would be welcome. Indeed, it behaves 'perfectly' with respect to a particular type of 'paradigmatic' simulated data.

We have modified this discussion after the addition of other simulated models (see Reply to Reviewer 1, above). We have modified this line to be less assertive.

L194. 'Confident calculation'. Which criterion is used to assess 'confidence'?

We refer here to how similar the recovery rates derived from AC1 and Variance are – if they are quite far apart, we would consider this a less reliable estimate. We have clarified this in text.

L209-2011. Here you propose an interpretation for problems in boreal forests – fine! Could you propose some explanation why close canopy tropical forest also raises problems? I agree that some 'issues' (L216 – note that no reference is provided) have been frequently evoked in literature but it could be good to recall briefly what issues seem important for the CSD-estimates topic.

Thank you for pointing this out. We refer here to well-documented saturation impacts of NDVI in dense vegetation, which especially leads to the large regions with low reliability for the tropical rainforests - where indeed several previous studies using MODIS optical indices had focused on. But this was not clearly stated in our original manuscript. We have updated this paragraph to be clearer.

L236-237. Some other areas seem to show increased 'resilience'. Why not mentioning and discussing them?

The main goal of our paper was to explore how and where resilience estimates can be considered reliable. Several other groups have published trends and discussed them in detail (Boulton et al., 2022, NCC; Feng et al., 2021 Communications Earth and Environment; Forzieri et al., 2022, Nature). We included trends in our manuscript for completeness, and to confirm that we still find a global tendency towards loss of resilience, as has been found previously (Smith et al., 2022 NCC). A full discussion of the spatial patterns of these trends is beyond the scope of our paper; hence we highlighted some clearly-defined regions of spatially coherent trends here.

We agree, however, that there are some relatively coherent regions in the southern United States and in south-eastern Africa that show increasing trends in the recovery rate - we now mention them as well in our Discussion.

L246-247. It was stated above that all MODIS indices poorly perform wrt dense vegetation (moreover also recalled in the next sentence). Why here advocating GPP so strongly?

The resilience of tropical forests has been explored in several papers (Verbesselt et al., 2016 NCC; Boulton et al. 2022 NCC, Forzieri et al. 2022 Nature), mostly using the NDVI, EVI, or kNDVI, which all perform very poorly; we hence wanted to mention the optical MODIS vegetation index that performed best in dense vegetation. To our knowledge, GPP has not so far been used in a discussion of CSD at the global scale. We, however, do not want to give the impression that GPP is well-suited to the analysis, merely that it is less bad than others which have been used recently, namely NDVI and kNDVI. We have modified this paragraph to be clearer.

L273. What was the script language used? Just specify to let the reader know without having to turn to Zenodo

We have added the language used here.

L322 and subsequent. There are some fundamental (and interesting) considerations provided here. For already established results some references should be provided (eg for Ornstein-Uhlenbeck processes) this is an ecology and evolution journal). Original results (if any) would deserve to be underlined (wrt eqs (4) to (8))

Thank you for highlighting this! We have added additional citations here, and clarified what has been shown before and what is new. Most of the equations and derivations are indeed known in mathematics (although we have not found the explicit form of Eq 8 anywhere else), but often not consistently applied in the natural sciences. We included them here to provide a firm theoretical underpinning to our analysis.

Decision Letter, first revision:

7th July 2023

Dear Dr. Smith,

Thank you for submitting your revised manuscript "Reliability of Vegetation Resilience Estimates Depends on Biomass Density" (NATECOLEVOL-23020369A). It has now been seen again by the original reviewers and their comments are below. The reviewers find that the paper has improved in revision, and therefore we'll be happy in principle to publish it in Nature Ecology & Evolution, pending minor revisions to satisfy the reviewers' final requests and to comply with our editorial and formatting guidelines.

[REDACTED]

Reviewer #1 (Remarks to the Author):

Thank you very much for your detailed response to my comments and suggestions. The extra analyses and insights are very interesting, well appreciated, and satisfactory answered my questions. I don't have any further comments or suggestions.

Our ref: NATECOLEVOL-23020369A

25th July 2023

26Dear Dr. Smith,

Thank you for your patience as we've prepared the guidelines for final submission of your Nature Ecology & Evolution manuscript, "Reliability of Vegetation Resilience Estimates Depends on Biomass Density" (NATECOLEVOL-23020369A). Please carefully follow the step-by-step instructions provided in the attached file, and add a response in each row of the table to indicate the changes that you have made. Please also check and comment on any additional marked-up edits we have proposed within the text. Ensuring that each point is addressed will help to ensure that your revised manuscript can be swiftly handed over to our production team.

****We would like to start working on your revised paper, with all of the requested files and forms, as soon as possible (preferably within two weeks). Please get in contact with us immediately if you anticipate it taking more than two weeks to submit these revised files.****

In recognition of the time and expertise our reviewers provide to Nature Ecology & Evolution's editorial process, we would like to formally acknowledge their contribution to the external peer review of your manuscript entitled "Reliability of Vegetation Resilience Estimates Depends on Biomass Density". For those reviewers who give their assent, we will be publishing their names alongside the published article.

Nature Ecology & Evolution offers a Transparent Peer Review option for new original research manuscripts submitted after December 1st, 2019. As part of this initiative, we encourage our authors to support increased transparency into the peer review process by agreeing to have the reviewer comments, author rebuttal letters, and editorial decision letters published as a Supplementary item. When you submit your final files please clearly state in your cover letter whether or not you would like to participate in this initiative. Please note that failure to state your preference will result in delays in accepting your manuscript for publication.

Cover suggestions

As you prepare your final files we encourage you to consider whether you have any images or illustrations that may be appropriate for use on the cover of Nature Ecology & Evolution.

Covers should be both aesthetically appealing and scientifically relevant, and should be supplied at the best quality available. Due to the prominence of these images, we do not generally select images

27featuring faces, children, text, graphs, schematic drawings, or collages on our covers.

Nature Ecology & Evolution has now transitioned to a unified Rights Collection system which will allow our Author Services team to quickly and easily collect the rights and permissions required to publish your work. Approximately 10 days after your paper is formally accepted, you will receive an email in providing you with a link to complete the grant of rights. If your paper is eligible for Open Access, our Author Services team will also be in touch regarding any additional information that may be required to arrange payment for your article.

Please note that *Nature Ecology & Evolution* is a Transformative Journal (TJ). Authors may publish their research with us through the traditional subscription access route or make their paper immediately open access through payment of an article-processing charge (APC). Authors will not be required to make a final decision about access to their article until it has been accepted. [Find out more about Transformative Journals](https://www.springernature.com/gp/open-research/transformative-journals)

Authors may need to take specific actions to achieve [compliance with funder and institutional open access mandates](https://www.springernature.com/gp/open-research/funding/policy-compliance-faqs). If your research is supported by a funder that requires immediate open access (e.g. according to [Plan S principles](https://www.springernature.com/gp/open-research/plan-s-compliance)) then you should select the gold OA route, and we will direct you to the compliant route where possible. For authors selecting the subscription publication route, the journal's standard licensing terms will need to be accepted, including [self-archiving-and-license-to-publish](https://www.nature.com/nature-portfolio/editorial-policies/self-archiving-and-license-to-publish). Those licensing terms will supersede any other terms that the author or any third party may assert apply to any version of the manuscript.

28[REDACTED]

[REDACTED]

Reviewer #1:

Remarks to the Author:

Thank you very much for your detailed response to my comments and suggestions. The extra analyses and insights are very interesting, well appreciated, and satisfactory answered my questions. I don't have any further comments or suggestions.

Author Rebuttal, first revision:

Reply to Reviewers – Smith and Boers – Nature Ecology and Evolution

Reviewer 1:

Thank you very much for your detailed response to my comments and suggestions. The extra analyses and insights are very interesting, well appreciated, and satisfactory answered my questions. I don't have any further comments or suggestions.

Thank you for taking the time to re-read our manuscript. We were happy to make the requested changes and further develop the manuscript after your valuable suggestions.

Final Decision Letter:

8th August 2023

Dear Dr Smith,

We are pleased to inform you that your Article entitled "Reliability of vegetation resilience estimates depends on biomass density", has now been accepted for publication in Nature Ecology & Evolution.

Over the next few weeks, your paper will be copyedited to ensure that it conforms to Nature Ecology and Evolution style. Once your paper is typeset, you will receive an email with a link to choose the appropriate publishing options for your paper and our Author Services team will be in touch regarding any additional information that may be required

Due to the importance of these deadlines, we ask you please us know now whether you will be difficult to contact over the next month. If this is the case, we ask you provide us with the contact information (email, phone and fax) of someone who will be able to check the proofs on your behalf, and who will be available to address any last-minute problems . Once your paper has been scheduled for online publication, the Nature press office will be in touch to confirm the details.

Acceptance of your manuscript is conditional on all authors' agreement with our publication policies (see www.nature.com/authors/policies/index.html). In particular your manuscript must not be published elsewhere and there must be no announcement of the work to any media outlet until the publication date (the day on which it is uploaded onto our web site).

Please note that *Nature Ecology & Evolution* is a Transformative Journal (TJ). Authors may publish their research with us through the traditional subscription access route or make their paper immediately open access through payment of an article-processing charge (APC). Authors will not be required to make a final decision about access to their article until it has been accepted. [Find out more about Transformative Journals](https://www.springernature.com/gp/open-research/transformative-journals)

Authors may need to take specific actions to achieve [compliance with funder and institutional open access mandates](https://www.springernature.com/gp/open-research/funding/policy-compliance-faqs). If your research is supported by a funder that requires immediate open access (e.g. according to [Plan S principles](https://www.springernature.com/gp/open-research/plan-s-compliance)) then you should select the gold OA route, and we will direct you to the compliant route where possible. For authors selecting the subscription publication route, the journal's standard licensing

31terms will need to be accepted, including <https://www.nature.com/nature-portfolio/editorial-policies/self-archiving-and-license-to-publish>. Those licensing terms will supersede any other terms that the author or any third party may assert apply to any version of the manuscript.

We welcome the submission of potential cover material (including a short caption of around 40 words) related to your manuscript; suggestions should be sent to Nature Ecology & Evolution as electronic files (the image should be 300 dpi at 210 x 297 mm in either TIFF or JPEG format). Please note that such pictures should be selected more for their aesthetic appeal than for their scientific content, and that colour images work better than black and white or grayscale images. Please do not try to design a cover with the Nature Ecology & Evolution logo etc., and please do not submit composites of images related to your work. I am sure you will understand that we cannot make any promise as to whether any of your suggestions might be selected for the cover of the journal.

You can generate the link yourself when you receive your article DOI by entering it here: <http://authors.springernature.com/share>.

[REDACTED]

P.S. Click on the following link if you would like to recommend Nature Ecology & Evolution to your librarian <http://www.nature.com/subscriptions/recommend.html#forms>

** Visit the Springer Nature Editorial and Publishing website at http://editorial-jobs.springernature.com?utm_source=ejp_NEcoE_email&utm_medium=ejp_NEcoE_email&utm_campaign=ejp_NEcoE for more information about our career opportunities. If you have any questions please click [here](mailto:editorial.publishing.jobs@springernature.com).**